# Asymptomatic *Bordetella pertussis* infections in a longitudinal cohort of young African infants and their mothers

Christopher J Gill[1†]*, Christian E Gunning[2†]*, William B MacLeod[1],
Lawrence Mwananyanda[1,3], Donald M Thea[1], Rachel C Pieciak[1],
Geoffrey Kwenda[4], Zacharia Mupila[3], Pejman Rohani[2,5,6]

[1]Boston University School of Public Health, Department of Global Health, Boston,
United States; [2]University of Georgia, Odum School of Ecology, Athens, Georgia;
[3]Right to Care, Lusaka, Zambia; [4]University of Zambia, School of Health Sciences,
Department of Biomedical Science, Lusaka, Zambia; [5]University of Georgia, Center
for the Ecology of Infectious Diseases, Athens, Georgia; [6]University of Georgia,
Department of Infectious Diseases, Athens, Georgia

**Abstract** Recent pertussis resurgence in numerous countries may be driven by asymptomatic infections. Most pertussis surveillance studies are cross-sectional and cannot distinguish asymptomatic from pre-symptomatic infections. Longitudinal surveillance could overcome this barrier, providing more information about the true burden of pertussis at the population level. Here we analyze 17,442 nasopharyngeal samples from a longitudinal cohort of 1320 Zambian mother/infant pairs. Our analysis has two elements. First, we demonstrate that the full range of IS481 qPCR CT values provides insight into pertussis epidemiology, showing concordance of low and high CT results over time, within mother/infant pairs, and in relation to symptomatology. Second, we exploit these full-range qPCR data to demonstrate a high incidence of asymptomatic pertussis, including among infants. Our results demonstrate a wider burden of pertussis infection than we anticipated in this population, and expose key limitations of threshold-based interpretation of qPCR results in infectious disease surveillance.

*For correspondence:
cgill@bu.edu (CJG);
research@x14n.org (CEG)

†These authors contributed
equally to this work

Competing interests: The
authors declare that no
competing interests exist.

Reviewing editor: Amy
Wesolowski, Johns Hopkins
Bloomberg School of Public
Health, United States

## Introduction

*Bordetella pertussis* remains a significant cause of morbidity and mortality among infants and young children around the world (*Yeung et al., 2017*; *Rohani and Scarpino, 2019*) and has experienced a resurgence in numerous countries despite long-standing vaccination programs (*Jackson and Rohani, 2014*; *He and Mertsola, 2008*; *Domenech de Cellès et al., 2016*; *Rohani and Drake, 2011*). Transmission by asymptomatic individuals is a suspected driver of pertussis resurgence (*Althouse and Scarpino, 2015*; *Cherry, 2013*) though unequivocal evidence documenting asymptomatic infections in adults and children is lacking. Rapid and reliable molecular diagnosis of pertussis is now possible using quantitative PCR (qPCR), which has supplanted microbiologic culture as the preferred tool for detecting pertussis (*Vincart et al., 2007*). Yet most pertussis surveillance uses cross-sectional monitoring that only captures a single point in time and thus cannot distinguish asymptomatic from pre-symptomatic infections.

In principle, repeated sampling over time could unambiguously identify asymptomatic infections. For example, if one could identify a patient at the moment of pertussis exposure and then regularly monitor them over time, one could anticipate capturing a gradual 'fade-in/fade-out' sequence as bacterial load, and thus qPCR signal intensity, varies across the arc of the infection: at initial exposure (where bacterial density is below the assay's limits of detection), during acute infection (as

bacterial density—and signal intensity—rises to a peak), and finally during recovery and convalescence (where an eventual loss of signal indicates pathogen clearance). Indeed, recent human infection trials have borne out this scenario (*Chilengi et al., 2020*; *DeVincenzo et al., 2020*). By sampling a single point in time, however, cross-sectional observations lack the historical context necessary to trace this arc of infection.

An additional complication arises when diagnostic thresholds are applied to qPCR cycle threshold (CT) values to distinguish positive and negative cases. For example, an IS481 CT <35 has been used as a diagnostic threshold of pertussis (*Tatti et al., 2011*; *Tatti et al., 2008*). This process introduces several problems, including calibration (e.g., between labs, machines, or over time) and a poorly examined trade-off between sensitivity and specificity that may also be task-dependent (e.g., clinical diagnosis). Further, this process discards information about qPCR signal intensity, such that borderline or low-intensity signals are summarily discounted as false positives or 'indeterminate'. These complications become particularly important as we extend the use of qPCR from clinical diagnosis into disease surveillance of populations (*Bolotin et al., 2018*).

Here we present an analysis of 17,442 nasopharyngeal samples (and associated IS481 qPCR assays) collected from 1320 Zambian mother/infant pairs who each provided at least four samples during the study. We begin with a descriptive analysis of eight mother/infant pairs where each symptomatic infant had definitive qPCR-based evidence of pertussis infection. We document the time course of infection in these individuals and observe frequent contemporaneous subclinical infections in the mothers of these infected infants. We then turn our attention to the entire cohort, where we use full-range IS481 CT values to show that qPCR signals of different intensities cluster in time, and we summarize within-subject variations in signal intensity over time. We then quantify the evidence for pertussis infection (EFI) within each individual across the full study. We show that EFI clusters within mother/infant pairs and is associated with clinical symptomatology and antibiotic use. We use these results to estimate the proportion of mothers and infants with evidence of asymptomatic, minimally symptomatic, and moderately/severely symptomatic pertussis in this cohort.

In total, we find that full-range CT values yield valuable insights into pertussis epidemiology in this population. Critically, the burden of pertussis here is substantially underestimated when restricting diagnostic criteria to IS481 CT ≤35. We also demonstrate widespread asymptomatic pertussis infections among mothers and, surprisingly, among young infants.

## Results

### Study overview

In 2015, we partnered with the Bill and Melinda Gates Foundation to conduct a prospective cohort study in Lusaka, Zambia (*Gill et al., 2016*). Between April and November 2015, we enrolled 1981 healthy Zambian mother/infant pairs (3962 individuals) shortly after birth and observed them during an additional six clinic visits scheduled at roughly 2–3 week intervals through approximately 14 weeks of age (when the last of the three routine infant DTP vaccine visits occurs). At each visit, we systematically obtained nasopharyngeal (NP) swabs and assessed symptoms and antibiotic use.

Of the initial cohort of 1981 pairs, 1497 mother/infant pairs attended at least one post-enrollment clinic visit, and 834 pairs attended all seven scheduled visits (including enrollment, *Table 1*). In this analysis, we focus on the 1320 pairs with ≥4 NP samples per subject (*Figure 1*). Baseline cohort demographics are shown in *Table 2*. Infants were enrolled at a median of 7 days post-partum; 47% were female, with a median gestational age of 40 weeks and birth weight of 3000 g. Mothers' median age was 25 years; >90% were married, and 17.5% were known to be infected with HIV. Among the HIV-positive mothers, nearly all were on antiretroviral therapy (ART) at the time of enrollment, and half had initiated ART prior to conception. Nearly all mothers received at least one dose of tetanus toxoid during pregnancy, signaling that some antenatal care was received by at least 99.5% of the maternal cohort. The final HIV status of the exposed infants could not be assessed.

### Descriptive analysis of eight noteworthy mother/infant pairs

In contrast to cross-sectional studies, our longitudinal analysis reveals the trajectory of qPCR signal intensity within subjects over time. To avoid confusion, we use the term 'detecting assay' throughout

**Table 1.** Study profile of cohort enrollment and attendance (bold indicates analysis set). Beyond eligibility and initial screening, the sole cause of cohort attrition was failure to attend one or more scheduled clinic visits. For eligibility and enrollment details, please see *Gill et al., 2016*.

| Study Phasephase | Mother/Infant Pairs |
|---|---|
| Recruitment and screening | 3033 |
| Initial enrollment | 1981 |
| Post-enrollment attendence | 1497 |
| **≥4 NP samples per subject** | **1320** |
| Attended all seven scheduled visits | 734 |

to describe a qPCR result with any detectable level of signal; all others are 'non-detecting' (N.D.). All qPCR assays were run for 45 cycles, making CT = 45 our effective limit of detection.

*Figure 2* provides a detailed timeline of IS481 CT results from the initial group of eight mother/infant pairs, which included all infants with a definitive positive NP sample (IS481 CT <35) during a clinic visit where respiratory symptoms were reported. This figure highlights the experimental design, where pairs were monitored across the infant's first months of life at regularly spaced clinic visits (e.g., pairs C and E), with additional mother-initiated visits for acute health care (e.g., pairs A and D). Minimal symptoms (cough and/or coryza) were common in both mothers and infants, while moderate to severe symptoms were much rarer, particularly among the mothers. Also rare is ptxS1 corroboration of IS481 results (pair D and infants F-H). This is not surprising, given the high IS481 copy number relative to the single ptxS1 copy, which renders the latter target specific but insensitive for detection of the bacterium (*Reischl et al., 2001*).

Of particular note, seven of these eight mothers had at least one detecting assay, while six had multiple detections. However, many of these mothers' detecting assays showed relatively weak signals (IS481 CT >40, e.g., mothers A, C, and H), with only a few results meeting CDC's recommended diagnostic criteria for a positive test (*Lievano et al., 2002*). We note, however, that CDC criteria were designed for application by clinicians when evaluating patients with severe and/or classic pertussis symptoms, and were intended to favor specificity over sensitivity. They were not designed for surveillance of asymptomatic individuals. Moreover, in our data these weak qPCR signals frequently bracketed visits with definitive test results and were observed in all eight infants who met CDC criteria at one or more visit.

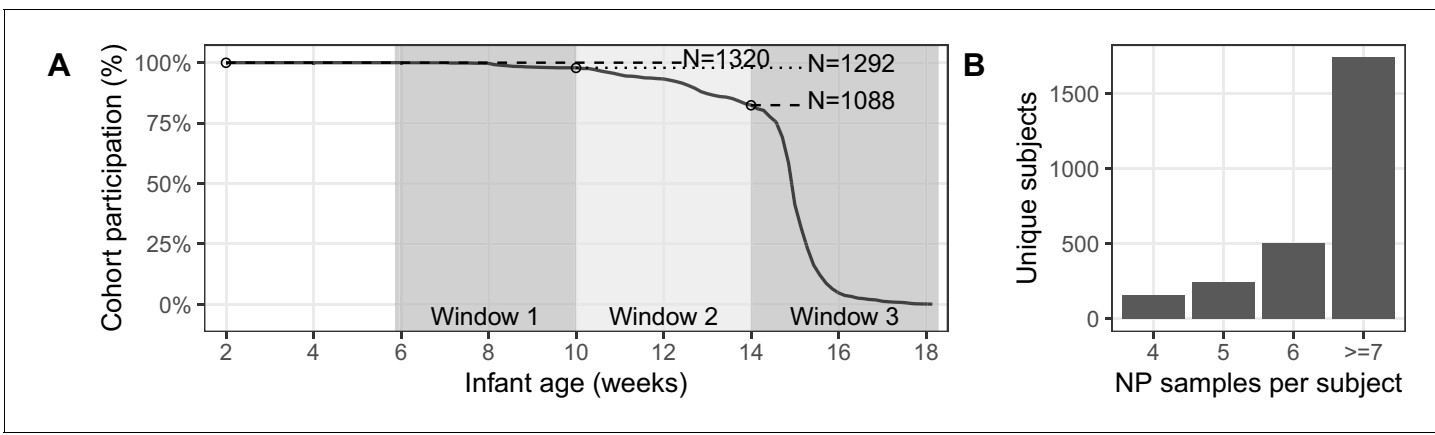

**Figure 1.** Study Attendance. (**A**) Percent attendance (%) of mother/infant pairs by infant age at last attendance (N = 1320, excluding pairs where subjects had <4 NP samples). Shaded regions show target age windows of DTP doses 1–3. Horizontal lines and text shows number of pairs attending up to marked ages: beginning of study enrollment, and at earliest timely administration of DTP doses 1–3. See *Table 1* for study profile. Most pairs (734/1320) attended all seven scheduled visits (including enrollment). (**B**) NP samples per subject: number of subjects with each sample count (including enrollment and unscheduled visits). Note that, with rare exceptions, each mother has the same number of NP samples as their infant.

**Table 2.** Demographic characteristics of participants (interquartile range in parentheses). Only subjects with at least four NP samples were included in subsequent analyses.

| Parameter | Study Participation | |
| --- | --- | --- |
| | Enrolled | ≥4 NP Samples |
| Number Under Study | 1981 | 1320 |
| Mothers | | |
| Married | 90.2% | 89.8% |
| HIV+ | 17.5% | 19.5% |
| Median Age | 25 (21, 29) | 25 (22, 30) |
| Median Infants In House (<1 year) | 1 (1, 1) | 1 (1, 1) |
| Median Children In House (<5 years) | 2 (1, 2) | 2 (1, 2) |
| Infants | | |
| Born at Chawama PHC | 56.9% | 56.6% |
| Born at UTH | 34.8% | 35.5% |
| Female sex | 46.9% | 46.1% |
| Median birth weight (kg) | 3 (2.8, 3.3) | 3 (2.8, 3.3) |

A clear example of pertussis infection fade-in and fade-out is provided by infant G, including ptxS1 confirmation and severe symptoms that occur after the observed IS481 signal peak. Remarkably, a weak (and asymptomatic) IS481 signal in infant G at age six weeks precedes the acute infection observed around age 8 weeks, while a weak IS481 signal in mother G is observed even earlier, before 4 weeks of age. This example suggests the likely sequence of transmission events within this mother/infant pair and thus influenced our emerging conclusion that weaker IS481 signals should not be automatically discounted.

Mother H provides another illustrative example, where seven of nine assays (78%) detect IS481, but none reach the canonical diagnostic threshold of <35, and none had a detectable ptxS1 result. A priori, the probability that all these detections were false positives appears low. In the context of an infected infant, this explanation becomes even less likely. While low sample pathogen density could also reflect clinically uninformative variation in sample collection and processing, the more parsimonious interpretation is that these seven NP samples contained pertussis, albeit at a low density.

While all infants in this first analysis were selected based on their presentation of symptomatic pertussis, several of their mothers presented with no respiratory symptoms at clinic visits. Of particular note are mothers G and H, each of whom had multiple detecting assays, strongly suggesting asymptomatic (or minimally symptomatic) pertussis infections.

Based on our initial results, we next assessed how frequent low-intensity signals were in our cohort by randomly selecting 500 NP samples from our catalogue of over 9000 maternal samples. None of these samples yielded detectable IS481. Applying the binomial theorem for an expected frequency of just under 1/500 (i.e., assuming that the next sample would have been detecting), the probability that seven of eight mothers of infected infants also had one or more detecting assays occurring by chance was <0.0001. We conclude that random chance is unlikely to account for the high concordance within these pairs, or for their evident tendency to coincide in time.

## Quantitative analysis of the full cohort

The low-intensity IS481 CT values (i.e., ≥35) discussed above would be adjudicated as 'negative' or 'indeterminate' in a typical cross-sectional study. However, we observed multiple lines of evidence supporting their microbiological and epidemiological significance that compelled us toward a comprehensive analysis of full-range IS481 CT values for the entire cohort.

## Concordance of qPCR signals over time

In *Table 3*, we summarize IS481 qPCR assays from the full cohort (17,442 NP samples total). Approximately 91% of all tests were non-detecting (N.D.), with 1561 detecting assays, including 818 in

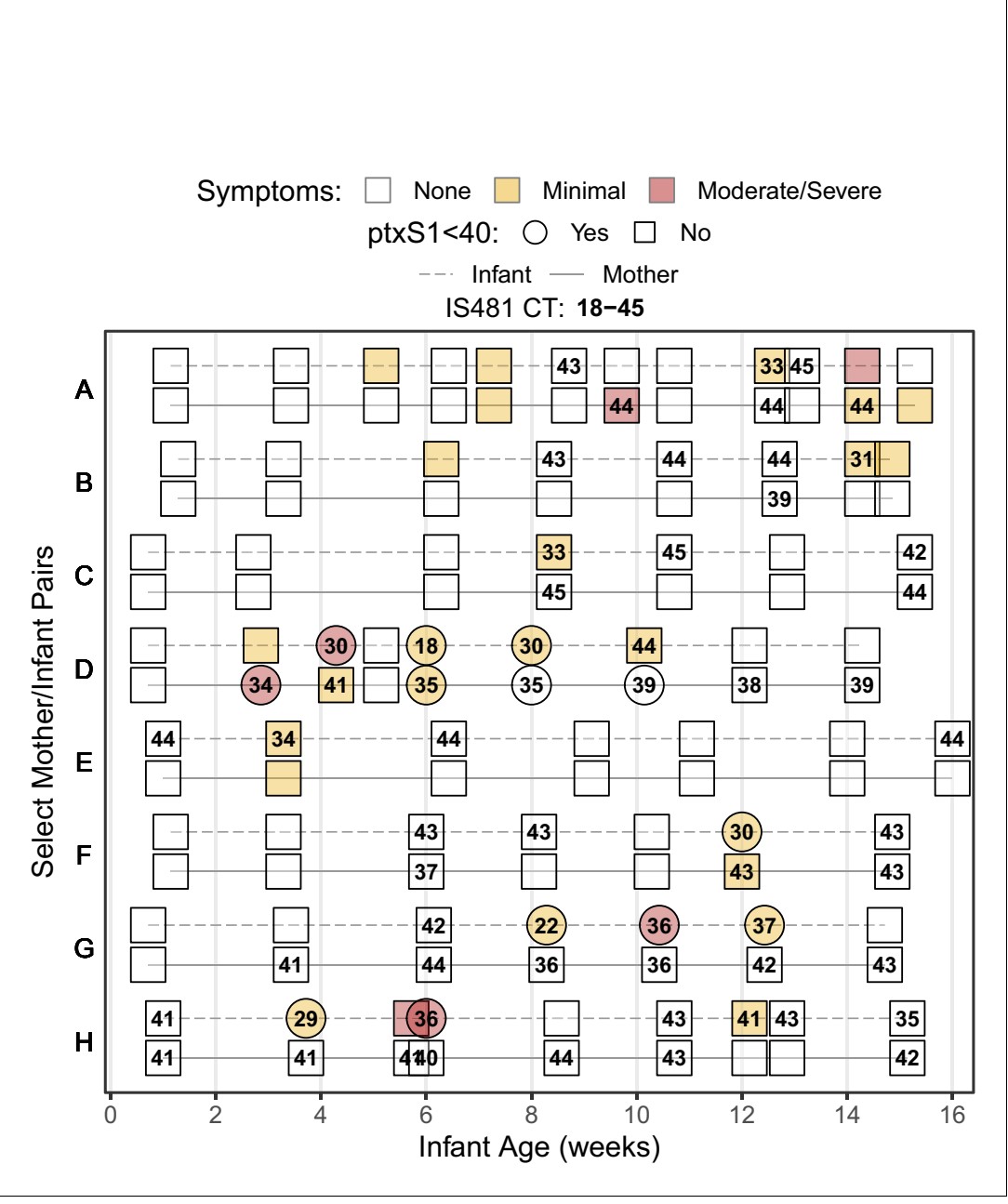

**Figure 2.** Timeline of study participation for eight noteworthy mother/infant pairs, showing rounded IS481 CT values (numbers), ptxS1 results (shape), and pertussis symptoms (color) at each clinic visit. Selected pairs include all symptomatic infants with definitive evidence of pertussis infection (IS481 CT <35). Blank cells show NP samples with no detected IS481. Contemporaneous detection of IS481 within pairs is common, as are temporal clusters of IS481 within individuals. Pertussis symptoms are relatively uncommon in mothers: of the seven mothers shown here with detectable IS481, four lacked any observable pertussis symptoms during clinic visits.

mothers and 743 in infants. Only 0.11% and 0.18% of mother and infant samples, respectively, had CT <35 and would have been considered definitive positive samples; all other samples would have been deemed indeterminate or negative based on traditional cut-points.

In *Figure 3*, we show the cohort's structure and time course. *Figure 3A* illustrates subject participation over calendar time for several example pairs that were chosen to highlight the cohort's rolling enrollment across 2015.

**Table 3.** Frequency of NP samples in each IS481 CT intensity stratum for infants and mothers (not detected: N.D.).

IS481 was detected in 1561 (8.95%) samples (743 in infants, 818 in mothers). Very few samples had CT <35: 16 samples (infants) and 10 samples (mothers).

| CT Strata | Infant | Mother | Sum |
|---|---|---|---|
| (18,40) | 99 (1.1%) | 60 (0.69%) | 159 |
| (40,43) | 254 (2.9%) | 276 (3.2%) | 530 |
| (43,45) | 390 (4.5%) | 482 (5.5%) | 872 |
| N.D. | 7980 (91%) | 7901 (91%) | 15,881 |
| Sum | 8723 (100%) | 8719 (100%) | 17,442 |

If weak qPCR signals (e.g., CT≥40) represent random background noise (i.e., false positives), then we would anticipate random variation in their frequency over time, and no evidence of correlation with stronger signals. To test this hypothesis, we assess the relative frequency over time of detecting assays grouped into three strata of qPCR signal intensities. In *Figure 3B*, the time-averaged frequency of NP samples in each stratum is shown by a separate curve, expressed as the percent of NP samples collected during that time period. Each of the 1320 study participants was tested repeatedly, and thus could contribute one or more detecting assays to each curve. Of particular note, we find that the mid- and low-intensity strata (e.g., 40≤CT<43 and 43≤CT) both closely mirrored the highest intensity strata (e.g., CT<40) in both mothers and infants, with a peak in June and July and a long tail of decline in September and October. This peak also coincides with a cluster over very high intensity assays (CT<35) that we highlight as points, and a related cluster of definitive infant pertussis cases described in our published 2016 analysis (*Gill et al., 2016*).

We further explore the temporal correlation among CT strata in *Figure 4*, which provides a detailed view of the weekly percent of NP samples in the high- and mid-intensity strata (CT<40 and 40≤CT<43, respectively). Here, each point represents a single study week in 2015, while color denotes time. Overall, these two strata are highly correlated (infants, ρ=0.68; mothers, ρ=0.71), and they rise and fall in synchrony (maximum cross-correlation at lag=0). However, the mid-intensity stratum is more sensitive to low infection prevalence in both infants and mothers, revealing a pertussis outbreak in April 2015 2–3 weeks prior to the high-intensity strata.

We note that cohort size alone cannot explain these results, as the cohort's size reached a steady state in June of 2015 that was sustained through the end of December 2015 (*Figure 3C*). Rather, these results are consistent with a population-level 'fade-in/fade-out' dynamic, where multiple overlapping signals from single individuals (e.g., *Figure 3A*) sum to create these curves. *Table 3* also highlights the preponderance of detecting assays with low signal intensity. We note that the range of CT values appeared greater for infants than mothers, with twice as many with CT results below 35 (15 vs 8, RR 0.54 95% CI 0.2–1.3).

## Impact of infant vaccinations

During our study, infants received routine whole-cell pertussis (wP) vaccinations according to the Zambian schedule at approximately 6, 10, and 14 weeks of age (see also *Gunning et al., 2020*). In *Figure 5*, we explore the impact of prior vaccination (i.e., number of doses administered at least 14 days prior to NP sample collection, *Figure 5A*) on detecting assays (*Figure 5B*). Here we see a gradual increase in the percent of infant assays that are detecting with increasing infant age, though no such pattern is evident in mothers. We interpret this increase in detections as cumulative ongoing exposure to pertussis in infants in the early weeks of life that achieves an steady state at ~age 8–10 weeks. While it is tempting to ascribe the flattening of this prevalence curve as possible evidence for a vaccine effect, we find no statistical evidence of an effect of prior vaccine dose beyond the (correlated) effect of infant age (*Figure 5B*). We also note that the steady state of approximately 10% detecting assays reached by infants at age 8–10 weeks approximately equals that of mothers. Critically, we note that the ideal study to identify any such interaction between prior vaccine dose and pertussis detection would be a randomized controlled trial and that, in this observational study, our data leave this question largely unanswered.

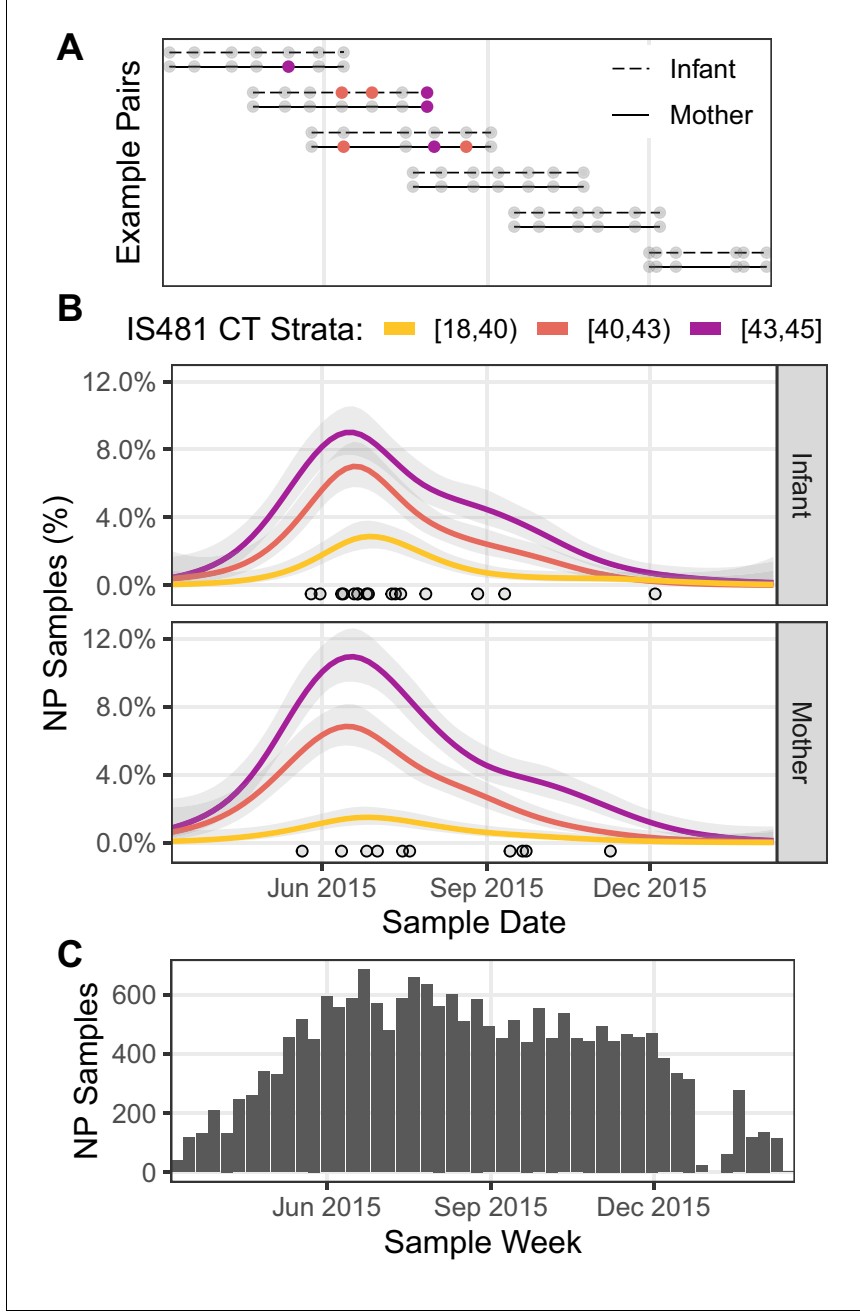

**Figure 3.** Timeline of study visits, NP samples, and IS481 assays. (**A**) Timeline of study participation for six mother/ infant pairs chosen to illustrate the cohort's rolling enrollment. Dots show clinic visits; color indicates NP sample IS481 CT strata. Visits included initial enrollment (shortly after birth) followed by (up to) six scheduled visits at 2–3 week intervals, and (in some cases) additional mother-initiated visits. (**B**) timeline NP samples for the full cohort (N = 17,442), showing the percent of samples with detectable IS481 over time, stratified by signal intensity (lower CT values indicate more IS481, see also *Table 3*). For each stratum, a generalized additive model estimated the time-varying proportion of all assays contained in that stratum (shading shows 95% CI). Points highlight assays with CT <35. A cluster of detecting assays in all strata peaks in late June/early July. Strong temporal correlation was observed among strata, and is consistent with detection of a pertussis outbreak, but is not consistent with randomly distributed false positive assays. (**C**) Number of NP samples per week (approx. denominator of B). The dip in Jan 2016 corresponds with the Christmas holiday.

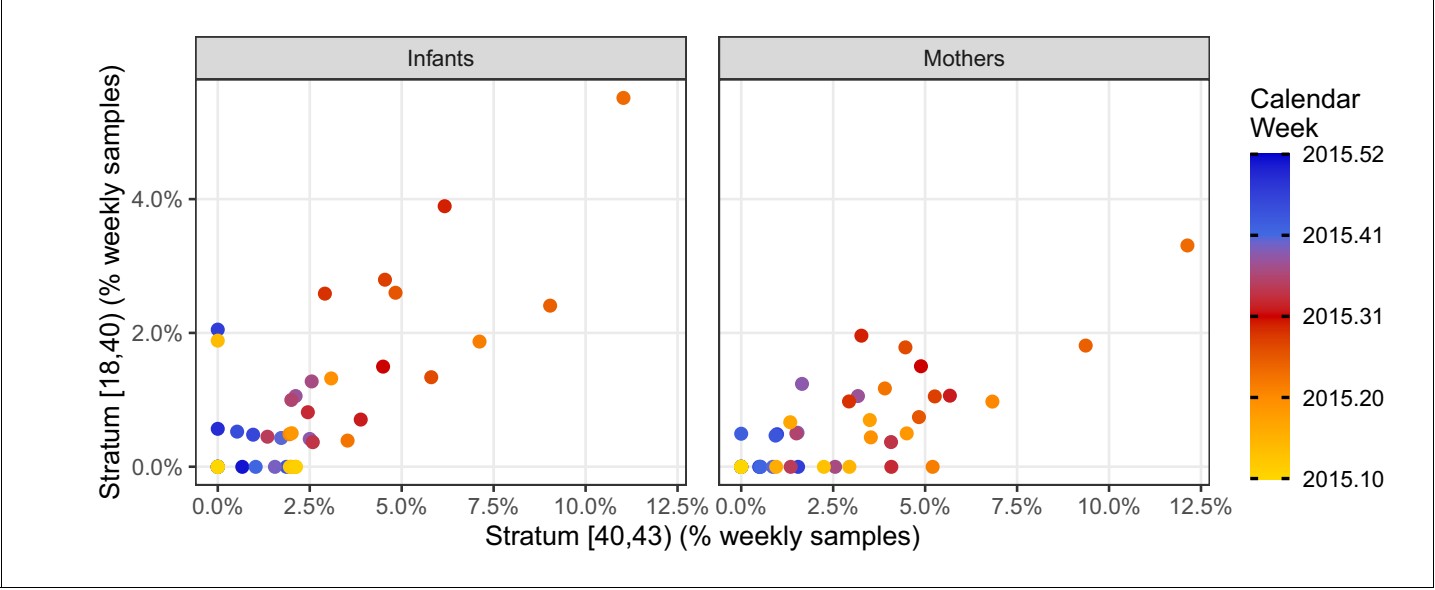

**Figure 4.** Phase portraits showing the weekly percent of NP samples in the mid-intensity stratum (X, 40 < CT $\leq$ 43) versus the high-intensity stratum (Y, CT <40) for infants (left) and mothers (right). Color shows calendar week (for clarity, weeks in 2016 are not shown). These strata are highly correlated: $\rho$ = 0.68 (infants); $\rho$ = 0.71 (mothers). The many weeks with Y = 0 and X > 0 illustrates the relatively low sensitivity of the high-intensity stratum. Indeed, the mid-intensity stratum detects a pertussis outbreak 2–3 weeks before the high-intensity stratum in April 2015 in both infants and mothers.

## Transitions of qPCR signals over time

We next explore the time-course of IS481 assays within individual subjects. If qPCR signals track the course of pertussis infection then, accepting the presence of random variation due to NP sample collection and handling and/or qPCR testing processes, we would nonetheless anticipate that adjacent samples would be more similar than dissimilar. For example, if a subject's first NP sample had CT = 44 (e.g., captured early in the infection process) then we would expect the next NP sample collected from this subject to be more similar (e.g., CT = 42) than different (e.g., CT <35.). Likewise, we expect fewer transitions from CT <40 to N.D. than from CT <40 to CT = 42. To explore this hypothesis, we conducted an analysis of pairwise transitions within individuals over time. In *Figure 6*, we summarize the relative frequency of transitions between qPCR signal intensity across adjacent NP samples (separated by no more than 25 days), where color shows the departure from expected frequency (assuming independence). Consistent with our hypothesis, we find that pairwise transitions tend to be cluster, with orderly transitions over time. In particular, transitions from detecting to detecting are much more common than expected by chance alone (red, lower left), while transitions from detecting to non-detecting are much less common (blue, right column) than expected. These results again demonstrate that full-range CT values contain epidemiologically relevant information consistent with an underlying biologic process.

## Quantifying evidence for pertussis infection

Our next analysis combines the contextual information provided by repeated sampling with that of full-range IS481 CT values to quantify the evidence for pertussis infection within individuals. As before, we focus on the 1320 mother/infant pairs where four or more NP samples (and associated IS481 assays) were available for each subject (see also *Figure 1B*).

We first summarize CT values across the study: we compute the reverse cumulative distribution (RCD) plots for CT values for mothers and infants separately (*Figure 7A*). We then use these RCDs to compute, for each subject, a summary statistic that we term the 'evidence for infection' (EFI): one minus the geometric mean RCD probability. Here, EFI=0 indicates no evidence (no detecting assays), while an EFI approaching 1 indicates very strong evidence arising from more detecting assays and/or stronger signals. We note that by averaging across time EFI provides no information about when infection occurred within the study. In *Figure 7B*, we show the distribution of EFI in mothers and

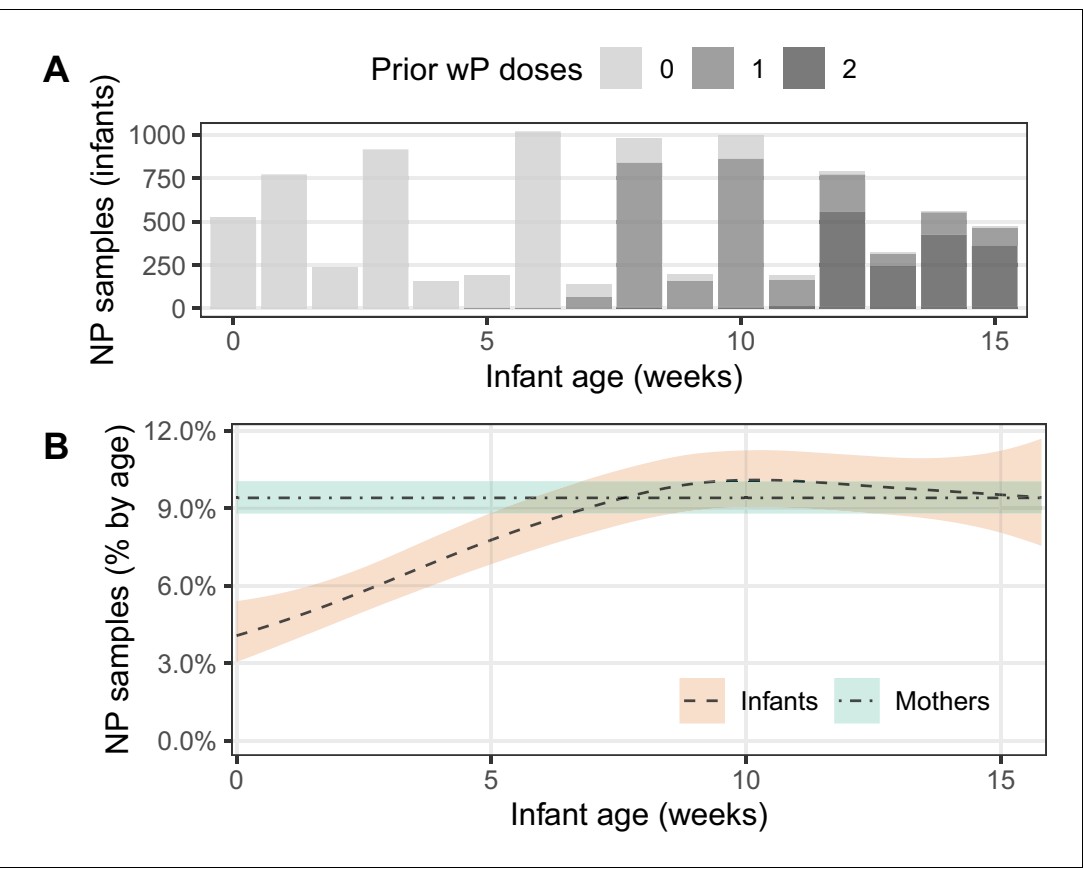

**Figure 5.** IS481 detections by infant age, showing wP vaccination schedule. (**A**) number of infant NP samples per week. Shading shows the number of wP doses received at least 14 days prior to sample collection. With rare exceptions, each infant sample is accompanied by a corresponding mother's sample. In most cases, the third wP dose was administered on the final study visit. (**B**) percent of NP samples with detectable IS481 over time with 95% CI (shading), estimated from generalized additive models (one each for mothers and infants). Infant age was a significant predictor of percent detection in infants only, while prior wP dose had no observable impact on percent detection in either infants or mothers.

infants stratified by the number of detecting assays. Under the premise that many false positives would be highly unlikely regardless of signal intensity, we categorize individuals with three or more detecting assays as having *strong EFI*, as well as other individual within this EFI range (dotted line, EFI≥0.52). Individuals with intermediate evidence (0<EFI<0.52) are categorized as having weak EFI.

## Concordance of evidence within mother/infant pairs

Multiple prior studies have reported that mothers and close family contacts of infected infants are very likely to also be infected (*Kara et al., 2017*; *Fedele et al., 2017*; *Skoff et al., 2015*), as consistent with our results in *Figure 2*. Here we use EFI to assess concordance of infection status within mother/infant pairs across the study. In *Figure 7C*, we show the intersection of EFI scores in each mother/infant pair as a single point. We use dotted and dashed lines (EFI > 0 and EFI > 0.52, respectively) to delineate possible EFI combinations for each mother/infant pair, for example mother-weak/infant-none or mother-strong/infant-strong. *Figure 7D* highlights the strong association of EFI category within mother/infant pairs. This association plot shows the frequency of concordance relative to expectation (assuming independent assortment), plotted as Pearson residuals. We find that when infants have no EFI, the corresponding mothers' EFI is also likely to be absent. Conversely, when infants display strong EFI, evidence in mothers also tends to be strong. We also repeat this analysis by varying the threshold defining a strong EFI score (using either two and four detecting assays, respectively), and find consistent results throughout (*Figure 7—figure supplement 1*). While this

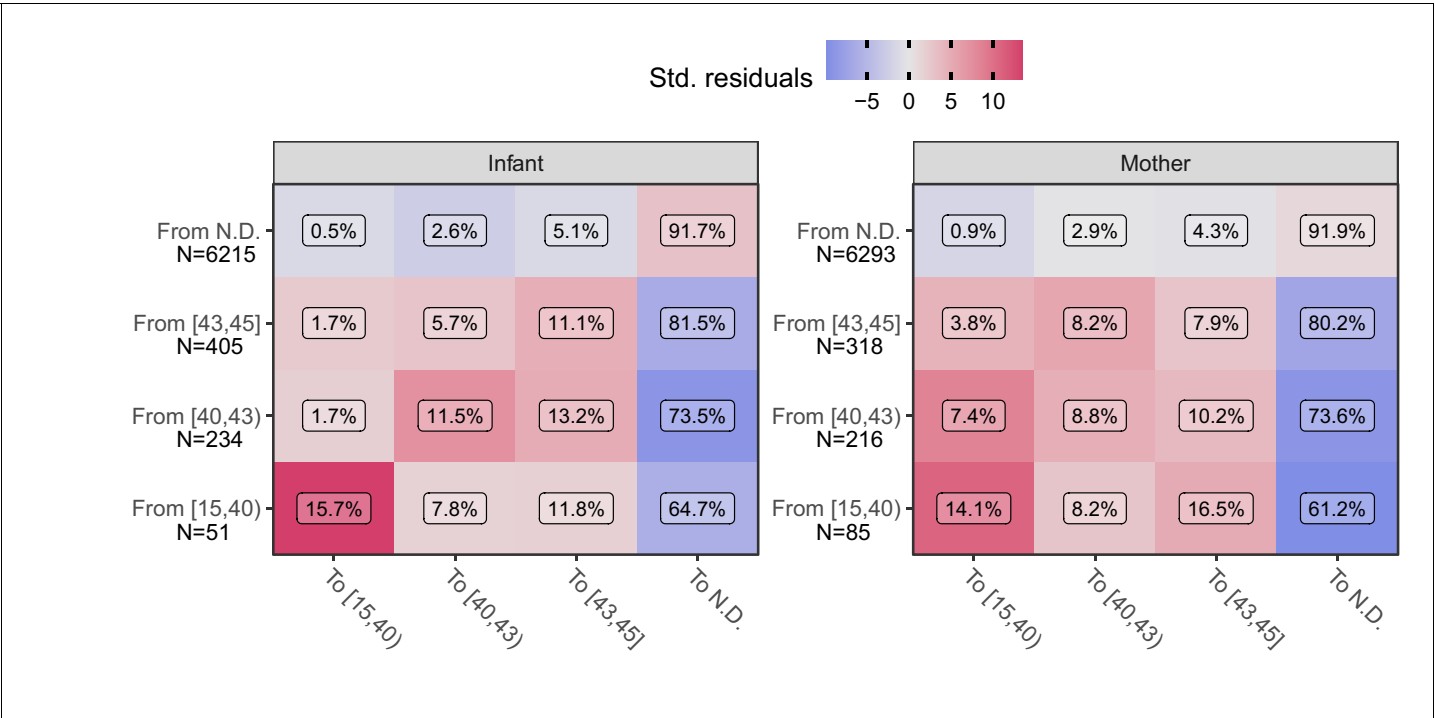

**Figure 6.** Transition frequency between IS481 CT strata over adjacent pairs of assays (within subjects) for infants (left) and mothers (right). Assay pairs separated by more than 25 days are omitted. N shows total transitions from each CT stratum (row); text shows percent of row total (N) within each cell. Assays were bootstrap resampled to generate a null distribution (1000 draws). Color shows standardized residuals: the difference between observed and expected frequency divided by the standard error of the difference. Transitions from detecting to detecting are more common than expected by chance alone (red), while transitions from detecting to non-detecting are much less frequent than expected (blue). See *Table 3* for marginal frequencies in each CT stratum.

analysis does not provide evidence of *contemporaneous* infection within pairs, these results offer strong evidence of transmission within the mother/infant pair.

## Concordance of evidence with symptoms

We expect a positive correlation between observed CT values and bacterial burden, and between bacterial burden and the presence and intensity of symptoms. Indeed, such a relationship has been shown for *Streptococcus pneumoniae*, where NP carriage density was much higher in children with pneumococcal pneumonia compared with asymptomatic carriers (*Piralam et al., 2020*; *Deloria Knoll et al., 2017*). As defined, our EFI uses each subject's full set of NP samples to provide an aggregate measure of the evidence for pertussis infection in that subject, and incorporates information regarding both the number of detecting assays and the CT values for each result. With this in mind, we tested whether EFI category was associated with cough and/or coryza (minimal symptoms), or additional pertussis symptoms (moderate to severe symptoms).

In *Table 4*, we tabulate the frequency of EFI category stratified by symptoms for mothers and infants. Focusing on those with the strongest molecular evidence for pertussis infections, we identified 188 mothers and 177 infants who met our criteria for a 'Strong EFI'. Within these groups, 130/188 (69%) of mothers had no symptoms, 55/188 (29%) had minimal symptoms, and only 3/188 (1.6%) had moderate to severe symptoms. By contrast, infants were more likely to be symptomatic than the mothers, but only in relative terms with 41/177 (23%) asymptomatic; 95/177 (54%) minimally symptomatic; and 41/177 (23%) with moderate to severe symptoms. Stated another way, 77% of infants with strong molecular evidence of infection were asymptomatic or minimally symptomatic. Only one infant presented with classic symptoms of whooping and paroxysmal cough. Such results are consistent with the theory that baseline host immune status (stronger in mothers, weaker or absent in infants) is at least partially responsible for mediating the clinical outcome of pertussis infections. It also emphasizes that minimally or asymptomatic infections occur commonly among infants.

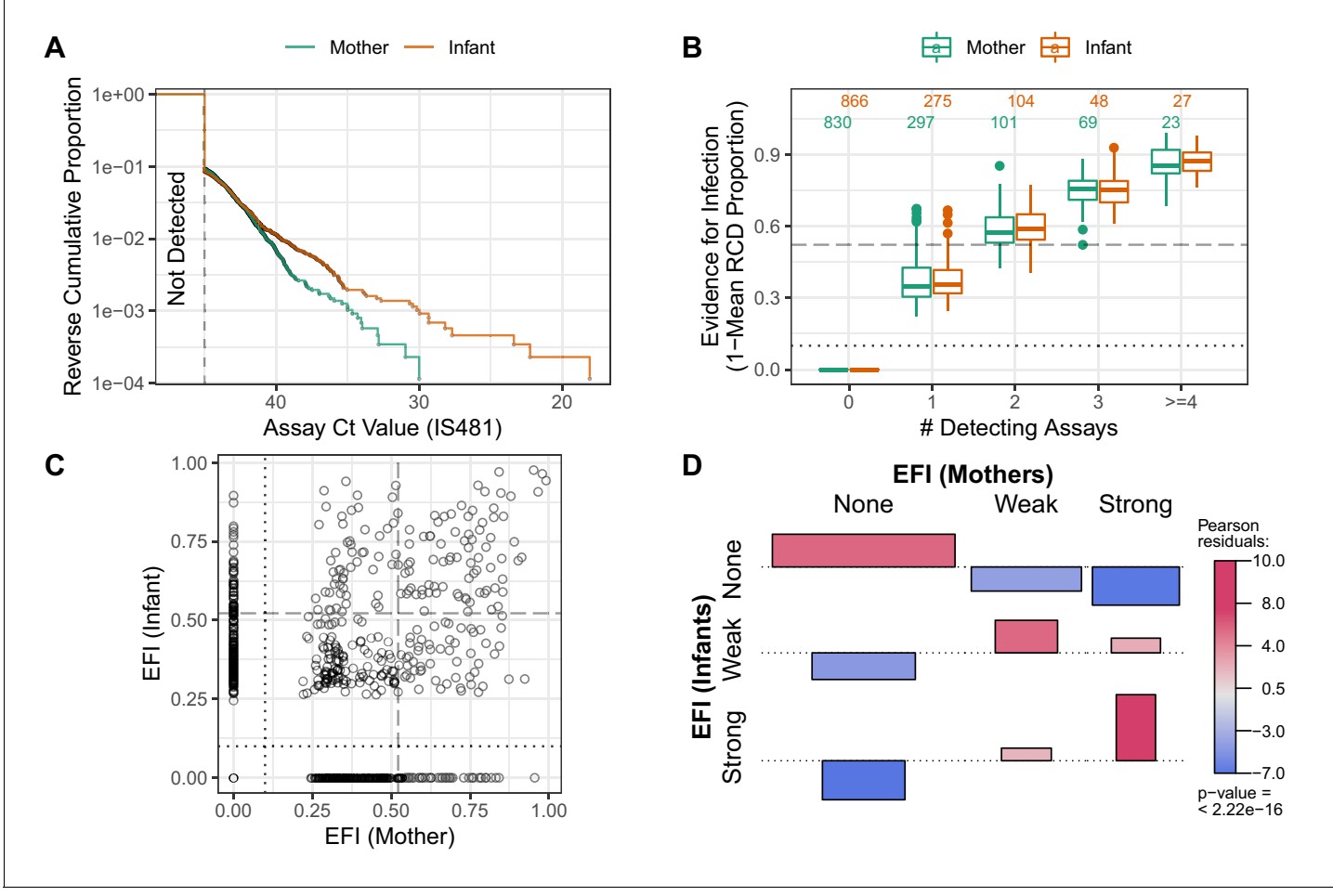

**Figure 7.** Quantifying evidence for pertussis infection, and concordance of evidence within mother/infant pairs. (A) Reverse cumulative distribution (RCD) curves of IS481 CT values for mothers and infants. (B) Boxplot summarizing evidence for infection (EFI), stratified by number of detecting assays per subject (x-axis). For each subject, EFI equals one minus the geometric mean RCD proportions (as in A). In general, EFI increases with lower CT values (A) and more detecting assays. The dashed line delineates strong evidence (defined to include all subjects with ≥3 detecting assays, 0.52 ≤ EFI < 1) from weak evidence (0 < EFI < 0.52; dotted line delineates no evidence (EFI = 0). (C) EFI in mother/infant pairs. Dotted and dashed lines as in B for mothers (vertical) and infants (horizontal). (D) Association of EFI strength (from C) between mothers and infants, showing very strong concordance (red) and rare discordance (blue) within pairs, particularly for pairs exhibiting strong EFI. Bar widths are proportional to expected counts; bar height and color show Pearson residuals (scaled difference between observed and expected counts). p-Value and residuals are relative to independent association. See also *Figure 7—figure supplement 1*.

The online version of this article includes the following figure supplement(s) for figure 7:

**Figure supplement 1.** As in *Figure 7*, varying the threshold of Strong EFI (dashed line) to include all individuals with ≥2, 3, or 4 detecting assays (A-C, respectively).

These data are represented graphically in *Figure 8A–B*, where color shows the frequency of each group relative to independent assortment. Among infants there is a strong positive relationship between EFI strength and the presence and severity of respiratory symptoms (*Figure 8A*). A similar but less pronounced pattern is observed in mothers, where moderate to severe symptoms were rare in mothers (*Figure 8B*). This relative absence of symptoms in mothers across infection status (i.e., for both weak and strong EFI) is consistent with the well-documented protective effects of pre-existing immunity (*Gill et al., 2017*). We also briefly summarize the association, within each clinic visit, between the presence of the most common symptoms (cough and/or coryza) and CT intensity strata in *Figure 8—figure supplement 1*. At this more granular scale, we find a similar pattern: in infants, symptoms are more commonly found with lower CT values and less frequently found with non-detecting assays (A), with a similar but much less pronounced trend in mothers (B).

**Table 4.** Frequency of EFI category in mothers and infants, stratified by occurrence of symptoms at any point during study participation.

Percentages are relative to row sums. Minimal symptoms include coryza and/or uncomplicated cough. Moderate to severe symptoms include all other pertussis symptoms in the Modified Preziosi Scale.

| | | EFI Category | | | |
| --- | --- | --- | --- | --- | --- |
| | Symptoms | None | Weak | Strong | Sum |
| Infants | None | 446 (77%) | 93 (16%) | 41 (7%) | 580 |
| | Minimal | 312 (58%) | 128 (24%) | 95 (18%) | 535 |
| | Moderate/Severe | 108 (53%) | 56 (27%) | 41 (20%) | 205 |
| | Sum | 866 (66%) | 277 (21%) | 177 (13%) | 1320 |
| | | | | | |
| Mothers | None | 669 (66%) | 209 (21%) | 130 (13%) | 1008 |
| | Minimal | 145 (51%) | 84 (30%) | 55 (19%) | 284 |
| | Moderate/Severe | 16 (57%) | 9 (32%) | 3 (11%) | 28 |
| | Sum | 830 (63%) | 302 (23%) | 188 (14%) | 1320 |

For illustrative purposes, we also provide in *Table 5* detailed results for each subject from the initial eight mother/infant pairs (i.e., from *Figure 2*), including total NP samples and percent detecting assays.

### Concordance of evidence with antibiotic use

Finally, we observe a strong positive association between EFI and antibiotic use in both infants and mothers (*Figure 8C–D*). We cannot infer a causal direction from the observed association. On the one hand, antibiotic use presumably serves as a proxy for symptoms at some prior time, since a clinician must have judged the individual sick enough to warrant treatment. On the other hand, antibiotic use has been shown to reduce the infectious dose of *B. pertussis* through effects on respiratory microbiota (*Zhang et al., 2019*). We also note that amoxicillin, which is not effective against *B. pertussis* but is a first-line therapy for empiric treatment of pneumonia, accounted for the vast majority of antibiotics use in this study. Erythromycin, which would be active against pertussis, is available in Zambia but is not part of the Integrated Management of Childhood Illness algorithm by which children are diagnosed with pneumonia, and hence rarely used. With the above in mind, our results call into question whether ineffective antibiotics have been commonly prescribed for mild pertussis cases in this (and possibly other) populations.

### Discussion

In this paper, we aim to accomplish two central goals. First, we aim to demonstrate that full-range IS481 CT values (and not just CT <35) offer valuable epidemiological insights, particularly when viewed in the context of repeated nasopharyngeal (NP) sampling of a prospective cohort of 1320 mother/infant pairs followed across the first 3 + months of life. To achieve this goal, we augment qPCR assays with clinical records, and leverage the temporal structure of the cohort across time to demonstrate the following: strong temporal correlation between assays of different CT strata (*Figure 3B*, *Figure 4*); evidence of stepwise transitions in CT values between adjacent NP samples within individuals (*Figure 6*); strong concordance of EFI within mother/infant pairs (*Figure 7D*); and a strong positive association between EFI and respiratory symptoms, particularly in infants (*Figure 8A*). Second, we aim to combine our findings into a coherent picture of pertussis epidemiology amongst infants in a low-resource urban setting. Here we find strong evidence that the true burden of asymptomatic and minimally symptomatic pertussis in this population is significantly under-reported, while individual infections are almost entirely untreated.

Our initial investigation of eight noteworthy mother/infant pairs suggested that full-range IS481 CT values contained useful information about the pertussis infection process. Building on this, we

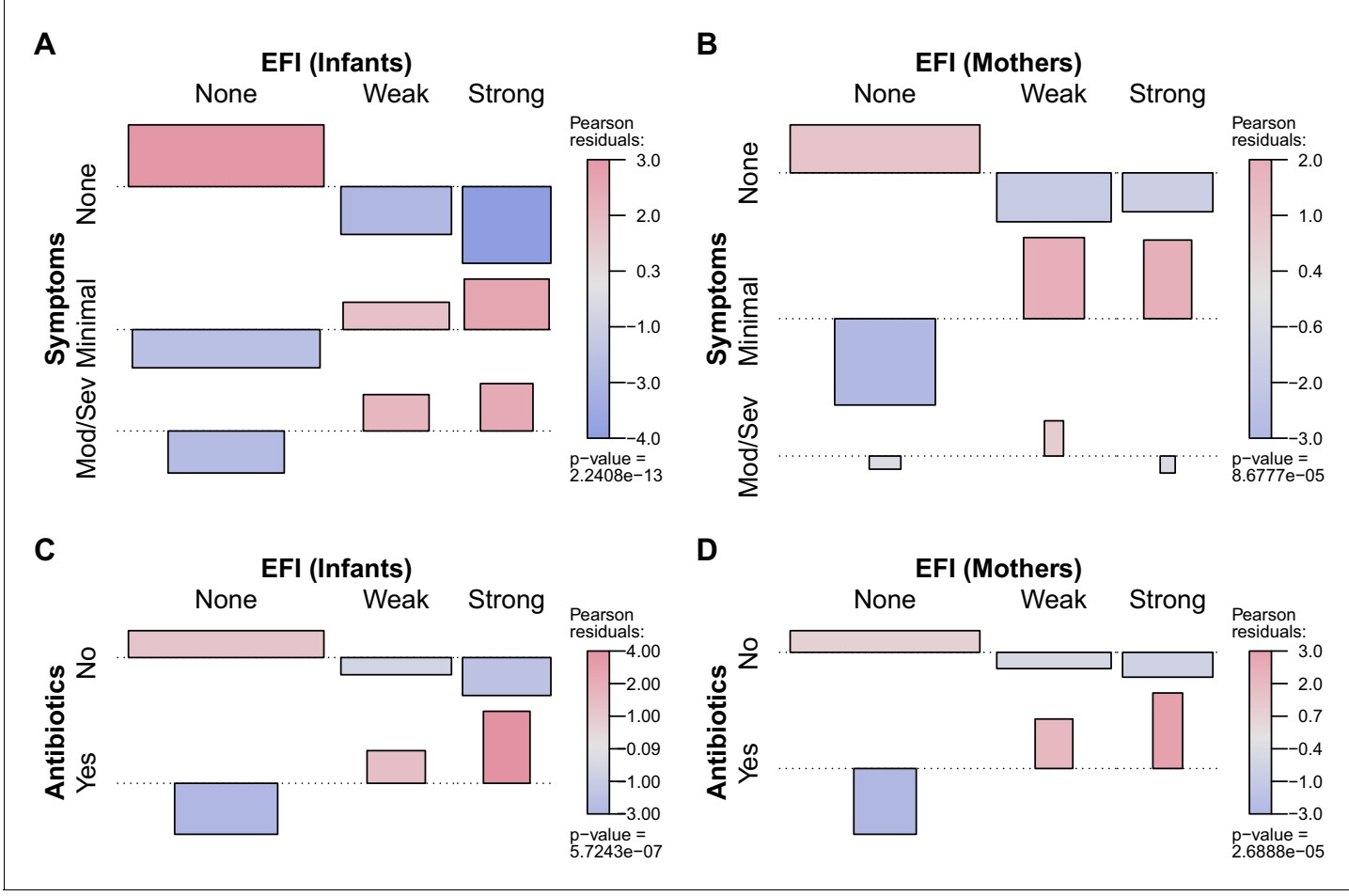

**Figure 8.** Association between participant EFI (as in *Figure 7*) and documented pertussis symptoms (A–B) or antibiotic use (C–D), separating mothers and infants (columns). Minimal symptoms include cough and/or coryza only; bar heights and p-values as in *Figure 7D*. In A-B, frequent co-occurrence of strong EFI with minimal symptoms is evident in both mothers and infants, as is no symptoms with no EFI. In infants, moderate to severe symptoms commonly co-occur with strong EFI, while more severe symptoms are rare in mothers. In C-D, frequent co-occurrence of antibiotic use with strong EFI is evident in both mothers and infants. See also *Table 4* and *Figure 8—figure supplement 1*.

The online version of this article includes the following figure supplement(s) for figure 8:

**Figure supplement 1.** Association between NP sample IS481 CT strata (columns, as in *Figure 3*) and presence of cough and/or coryza for A, infants, and B, mothers.

conducted a comprehensive analysis of our extensive library of NP samples and directly-observed clinical records that were prospectively collected from a representative African urban cohort of over 1300 Zambian mother and infants followed longitudinally. We find that the true burden of pertussis infection in this population was much higher than we expected, most of which reflected asymptomatic or minimally symptomatic infections. We also observe clustering of qPCR signals over time consistent with a local outbreak, clustering of detecting assays within mother/infant pairs, and a pattern of stepwise transitions in qPCR signal intensity among those with repeated detecting assays. Finally, we observe a strong association between the pertussis EFI and pertussis symptoms and the use of antibiotics. Collectively, these findings indicate that our longitudinal analysis of full-range, unthresholded qPCR data provided valuable and novel insights into the scope and scale of pertussis transmission in this population.

A key finding of our investigation is the common occurrence of asymptomatic (and/or minimally symptomatic) pertussis infections in adults. This issue has long been a subject of debate and speculation but has remained unresolved due to the lack of longitudinal data sets that would allow one to distinguish asymptomatic from pre-symptomatic infections (*Althouse and Scarpino, 2015*;

**Table 5.** Details for subjects shown in *Figure 2*, including NP sample number, percent samples with detected IS481, EFI, and summary of pertussis symptoms and antibiotics use.

| Subject | ID | Samples | Detected | EFI | Symptoms | Antibiotics |
|---------|-----|---------|------------|---------------|----------|-------------|
| A Infant | 126 | 11 | 3 (27.3%) | Strong (0.68) | Mod/Sev | Yes |
| A Mother | 126 | 11 | 3 (27.3%) | Strong (0.52) | Mod/Sev | Yes |
| B Infant | 269 | 7 | 4 (57.1%) | Strong (0.89) | Minimal | Yes |
| B Mother | 269 | 8 | 1 (12.5%) | Weak (0.50) | None | No |
| C Infant | 434 | 7 | 3 (42.9%) | Strong (0.84) | Minimal | No |
| C Mother | 434 | 7 | 2 (28.6%) | Weak (0.52) | None | No |
| D Infant | 474 | 9 | 4 (44.4%) | Strong (0.94) | Mod/Sev | Yes |
| D Mother | 474 | 9 | 7 (77.8%) | Strong (0.99) | Mod/Sev | Yes |
| E Infant | 573 | 7 | 4 (57.1%) | Strong (0.87) | Minimal | Yes |
| E Mother | 573 | 7 | 0 (0.0%) | None (0.00) | Minimal | No |
| F Infant | 579 | 7 | 4 (57.1%) | Strong (0.90) | Minimal | No |
| F Mother | 579 | 7 | 3 (42.9%) | Strong (0.85) | Minimal | No |
| G Infant | 691 | 7 | 4 (57.1%) | Strong (0.96) | Mod/Sev | Yes |
| G Mother | 691 | 7 | 6 (85.7%) | Strong (0.98) | None | No |
| H Infant | 752 | 9 | 7 (77.8%) | Strong (0.98) | Mod/Sev | Yes |
| H Mother | 752 | 9 | 7 (77.8%) | Strong (0.95) | None | No |

*Wiley et al., 2013*; *Craig et al., 2020*; *Tan et al., 2005*; *de Graaf et al., 2020*; *Rubin and Glazer, 2016*). A partial exception are a pair of sero-conversion studies in household contacts of index symptomatic pertussis infections in children. Both observed frequent seroconversions among individuals who, looking back over the prior year, could not recall a pertussis-like infection. Such data support the theory of asymptomatic pertussis infections, but were not definitive proof since they rested on possibly faulty recall of illnesses over a preceding year, and the fact that pertussis infections may be indistinguishable from common viral infections, particularly in adults, as the data in this analysis have emphasized (*Long et al., 1990*; *Deen et al., 1995*). Far more surprising, however, is the relatively high frequency of asymptomatic infections among infants. Pertussis in infants has been assumed to be predominantly severe and occasionally mild; our data indicate otherwise (*Tozzi et al., 2003*; *O'Riordan et al., 2014*; *Namachivayam et al., 2007*).

Older children, adolescents and young adults are generally recognized as the reservoir for pertussis, (*Mink et al., 1994*; *Heininger, 2010*; *Brooks and Clover, 2006*; *von König et al., 2002*) while mothers and older siblings are often the source of pertussis in infants (*Armangil et al., 2010*; *McIntyre and Wood, 2009*; *Schellekens et al., 2005*; *Bisgard et al., 2004*). While the relationship between qPCR signal intensity and infectiousness may be modified by other factors, including whether the individual is symptomatic or whether *B. pertussis* is growing freely or embedded within a biofilm matrix, (*Cattelan et al., 2017*) signal intensity nonetheless serves as a rough proxy for infectiousness (*Singanayagam et al., 2020*; *Shaikh et al., 2019*; *Mondal et al., 2019*; *Lau et al., 2010*). In our study, the distribution of IS481 CT values were similar for infants and mothers, raising the question of whether infants were merely the recipients of pertussis infections, or whether they could also participate in chains of transmission.

This analysis has several limitations. We have not provided a formal analysis of the related species *B. parapertussis* or *B. holmesii*, though initial estimates suggest that neither plays a major role in this system. Similarly, our assay does not distinguish between *B. pertussis* and *B. bronchiseptica*, an infection primarily of animals with occasional spill over into humans. This could lead to some degree of misclassification, though the effect is likely to be extremely small given how rarely this organism has been detected in human studies (*Van den Bossche et al., 2013*). We have not directly inferred pertussis transmission, and acknowledge that a linear relationship may not exist between qPCR signal intensity and infectiousness. In addition, several key questions fell outside of the scope of this analysis and warrant further exploration. Notably, a time-series analysis of the trajectory of qPCR

signals within individuals, and within mother/infant pairs, remains an important next step, as does a more nuanced estimation of the incidence of symptomatic and asymptomatic pertussis. We also acknowledge that ascribing symptom severity or absence is inherently subjective, and relies here on self-reporting across a time-scale of weeks. Nonetheless, symptoms were prospectively evaluated by trained clinic staff using a standardized questionnaire. Lastly, our analysis could not assess the impact of infant HIV infection on these relationships. While we had data on maternal HIV status, our study was not able to ascertain final HIV status on the infants themselves, which typically is not possible until the cessation of breast feeding. We assume that these HIV +infants would be rare, since all of the mothers had to be receiving antiretroviral therapy during pregnancy as an inclusion criterion for the study.

A key strength of our study is that our data were systematically collected according to a schedule rather than adventitiously sampled in response to symptoms. Such prospective surveillance ensures these data are robust against selection bias, which is the primary threat to any cohort. Moreover, as published previously about this cohort, we had a high retention rate, and the majority of retained participants attended all seven scheduled visits (*Gunning et al., 2020*). During the study, our team offered convenient access to participants' acute care needs, as well as travel stipends at each visit, thus reducing the likelihood that infection events were missed due to patient seeking care elsewhere. In addition, automated sample tracking allowed us to link qPCR results directly to concurrently measured respiratory symptoms and clinical data including antibiotic use, which greatly increased our confidence that we could sort symptomatic from asymptomatic infections.

We note that *B. pertussis* has proved to be a valuable study organism for several reasons. First, IS481 provides a very sensitive target due to its high copy number, while *B. pertussis* can be easily collected from the nasopharynx. In addition, pertussis infections can last for many weeks (or even months), permitting a sampling frequency on the order of once every few weeks. By contrast, the arc of most viral respiratory infections is measured in days (or a few weeks), and more frequent sampling would be required to observe these shorter arcs of infection. Nonetheless, the general principles outlined here remain salient: a weak signal tracked over time can provide valuable insight into biological processes.

Even in situations where repeated, longitudinal sampling of individuals is not feasible, a careful analysis of full-range qPCR data could be applied to repeated, prospective cross-sectional population sampling. Of particular note, we found that: (A) qPCR signals of different intensities were highly correlated in time but that, (B) high-intensity signals were much less frequently observed, again highlighting the intrinsic trade-off between sensitivity and specificity. Our results show that high-intensity qPCR signals were insensitive indicators of low disease prevalence, while mid-intensity signals were able to detect the early phase of a pertussis outbreak 2–3 weeks earlier. Operationally, the increased sensitivity offered by lower-intensity qPCR signals can also render smaller sample sizes more informative. Finally, an important outstanding question is whether a sustained population-level rise in low-intensity qPCR signals could provide early warning for epidemics of RSV, influenza, or CV19. Indeed, there is an emerging dialogue about the use of full-range qPCR data as tools in epidemiologic surveillance (*Hay JA et al., 2020*; *Haramoto et al., 2018*).

In conclusion, these results provide an important benchmark of the relative frequency of asymptomatic and minimally symptomatic pertussis infections in both adults and infants. More broadly, the longitudinal structure of our data suggests that the use of canonical diagnostic thresholds in qPCR (e.g., IS481 CT <35) has the unintended effect of removing valuable epidemiological information, which is particularly impactful of population-level surveillance. Our findings also have implications across a broad range of infectious disease surveillance efforts, where we believe a similar approach could provide earlier and more sensitive detection of infectious disease outbreaks.

## Materials and methods

### Study design

The Southern Africa Mother Infant Pertussis Study (SAMIPS) was a longitudinal cohort study conducted in Lusaka, Zambia that followed mother/infant pairs through the infants' first three months of life. To this end, we sought to enroll all healthy live births that occurred between March and December 2015 in Chawama compound, a densely populated peri-urban slum near central Lusaka. A

detailed account of study methods (including sample size considerations) was published previously (*Gill et al., 2016*).

Enrollment was conducted at the Chawama Primary Health Clinic (PHC), and mother/infant pairs were recruited during their first scheduled postpartum well-child visit at approximately 1 week of age. Chawama PHC is the only government-supported clinic in this community, and is the primary source of medical care for Chawama residents, allowing us to maximize study reach. Prior to study initiation, a public outreach campaign also provided study information to pregnant Chawama residents.

Infant enrollment eligibility included the following: full-term (infants were born after 37 weeks), birth weight >2500 grams, and delivered without complications or apparent disease. Maternal eligibility included signed consent, Chawama residency (anticipated remaining in the community during study period), known HIV status prior to delivery, and treatment with prophylactic antiretroviral therapy at the time of delivery for HIV +mothers.

Mothers were incentivized to join and remain in the cohort in three ways. First, the SAMIPS medical staff provided all routine and acute medical care for study participants during their time of enrollment. This significantly reduced waiting times for care from over 3 hr to half an hour or less. Second, mothers received a travel stipend for each visit valued at approximately 7 US dollars. Lastly, a small gift of baby supplies was provided for mothers attending the final scheduled study visit.

After the baseline enrollment visit, infants were scheduled for six routine follow-up clinic visits at 2–3 week intervals through approximately 14 weeks old (maximum, 18 weeks). Additional unscheduled clinic visits were initiated by study mothers for acute medical care as well as routine well-child care. At each clinic visit, nasopharyngeal (NP) swab samples were obtained from both mother and infant, and detailed records of current respiratory symptoms were collected on a standardized reporting sheet for each subject by clinic staff. Unique barcodes were assigned to study records from pre-printed sticker books, and were used to link subjects, clinic visit records, and NP samples. Each barcode was scanned at the time of assignment using the Xcallibre digital pen system. We note that qPCR results were unavailable during the study, and thus could not affect symptom assessments and clinical management decisions.

Routine childhood vaccinations were administered during scheduled clinic visits. Diphtheria-Tetanus-Pertussis (DTP) doses 1–3 were administered at visits corresponding to 6, 10, and 14 weeks of age as a pentavalent combination (Pentavac, Serum Institute of India Limited, Pune, India) that included whole-cell pertussis, *Haemophilus influenzae* type B (HIB), and Hepatitis B. The pneumococcal conjugate vaccine (PCV10) was co-administered with DTP, and the oral rotavirus vaccine administration was scheduled for 6 and 10 weeks of age. Additional details regarding infant vaccination are provided in *Gunning et al., 2020*. Routine childhood vaccinations were provided by regular clinic staff at no cost in a separate area of the clinic compound located ~50 feet from where the NP sampling was done. This was intended to avoid contamination of the swabs by pertussis DNA present within the pertussis vaccines themselves, a known cause of pseudo-outbreaks of pertussis in health care settings (*Mandal et al., 2012*).

## Nasopharyngeal sampling

NP samples were obtained using flocked-tipped nylon swabs (Copan Diagnostics, Merrieta, California) (*Van Horn et al., 2008*) that were inserted into each nostril until contact with the posterior nasopharynx was made. Swabs were then rotated 180 degrees in both directions, placed in commercially prepared tubes with universal transport media (UTM), and stored on ice until transport. Samples were collected from the study clinic daily and were taken to the PCR laboratory at the University Teaching Hospital (UTH), where they were stored at −80°C.

## Laboratory methods

NP sample DNA was extracted using the NucliSENS EasyMag system (bioMérieux, Marcy l'Etoile, France) (*Perandin et al., 2009*; *Kim et al., 2009*). Samples were initially tested for *B. pertussis* using a singleplex TaqMan qPCR genomic assay targeting the IS481 insertion sequence. In addition, a qPCR assay tested each sample for the constitutively expressed human RNase P (RNP) to assess successful sample collection, storage, DNA extraction, and lack of PCR inhibition. Each 96-well qPCR plate contained approximately 46 samples (one each of IS481 and RNP), along with a positive and

negative control for each assay. We note that a lower CT value indicates a greater quantity of target. Each reaction was run for 45 cycles, such that the minimum detectable target quantity has a CT value of 45. We consider assays with a CT value of 45 or less to be *detecting* assays; all others are *non-detecting* (N.D.) assays.

For the descriptive analysis of the first eight symptomatic infants and their mothers, an IS481 detection was followed by a second assay targeting the ptxS1, the gene for a sub-unit of pertussis toxin. Given the high volume of testing, we only used IS481 for the full analysis of the library. All primers and probes were purchased from Life Sciences Solutions (a subsidiary of ThermoFisher Scientific Inc). Most samples were run using an ABI 7500 thermocycler (ThermoFisher Scientific Inc, Waltham, MA). Starting in 2019 some samples were also run on a QuantStudio5 thermocycler (ThermoFisher Scientific Inc, Waltham, MA). An analysis of samples run in parallel on both machines showed minimal systematic variation between machines, so we do not distinguish these in the current analysis.

## Data and statistical analysis

### Descriptive analysis of the first eight mother/infant pairs

As presented in our 2016 paper, from the 1981 infants in SAMIPS, we initially selected infants presenting with any respiratory symptoms (rather than classic pertussis symptoms) for qPCR testing (*Gill et al., 2016*). Thus, any pertussis detected in these infants will be 'symptomatic pertussis' by definition. Given our focus at this stage on detecting symptomatic pertussis, we defined pertussis strictly as per the US CDC's protocol: any IS481 insertion sequence CT <35, or an IS481 of 35–40 plus a CT of <40 for ptxS1, the gene that codes for pertussis toxin. It should be noted that IS481 is a very sensitive probe as *B. pertussis* carries multiple gene copies per organism. By contrast, there is usually only one copy of ptxS1 per bacillus, making it highly specific but insensitive (*Register and Sanden, 2006*; *Qin et al., 2007*). We then expanded testing for IS481 and ptxS1 to all of the other samples within those eight mother/infant pairs.

### Systematic analysis of the full data set

We focus here on IS481 qPCR CT values. We did not include ptxS1 in these analyses due to its lack of sensitivity. While this introduces the possibility that some IS481 detections were due to a species other than *B. pertussis,* we expect a minimal impact of rare *B. pertussis* false positives on our findings. As the duration of study participation varied considerably between subjects (*Table 1*), we focus on subjects with at least 4 NP samples.

#### Temporal analysis

We first conduct an exploratory data analysis of IS481 CT values over time. We group assays into arbitrary ranges of CT values such that each range of decreasing CT value (greater target) contains fewer samples than the previous interval. We then use a set of generalized additive models (GAMs) to describe the relative frequency of samples in each CT range over the course of the study (binomial link function, smoothed by calendar date using cubic regression splines with shrinkage, one model per stratum). We use these models to visually compare the relative frequency and timing of IS481 signal intensity. We also use two GAMs (mothers and infants separately) to estimate the (smoothed) effect of infant age on expected percent of detecting assays (binomial link function). We then refit these models using the number of doses administered >14 days prior to NP sample collection as a categorical predictor, and used a Chi-Squared model comparison test to assess the additional explanatory value of vaccine dose.

#### Evidence for Infection

For each participant, we compute a summary measure of their IS481 CT values across the course of the study, which we refer to as the *evidence for infection* (EFI). To compute the EFI, we first compute the reverse cumulative distributions (RCD) of CT values over all samples in the study (for mothers and infants separately). Emphasizing that the CT is roughly equivalent to the inverse of pathogen density, from these RCDs, each IS481 assay is now associated with a probability describing its rarity in the study, ranging from 1 (not detected after 45 cycles) to 0 (lowest CT value in the study). For each subject, we then compute the geometric mean RCD probability of that subject's assays. One

minus this mean proportion yields the *evidence for infection* (EFI) in this subject during the study period. Conceptually, zero detecting assays (EFI = 0, all N.D.) indicates no evidence for infection, whereas an EFI approaching one indicates strong evidence for infection at some point during the study (but does not provide information about the timing of infection).

We also assess the number of detecting assays per subject. We consider individuals with three or more detecting assays as very likely to have experienced *Bordetella* infection during the study. We then identify the minimum EFI of these individuals, and take this value as a threshold delineating *strong* evidence for infection (EFI $\geq$ threshold). Note that, since our false detection rate is bounded above by approximately 10% (assuming all detections were false, *Table 3*), our false classification rate given three or more detecting assays must be less than $(10\%)^3$ = 0.1%. Subjects with EFI below this threshold but greater than zero we consider as having *weak* evidence for infection, while subjects with EFI equal to zero have no evidence for infection (all N.D.). Thus, each mother and infant in the analysis set is assigned into one of three mutually exclusive categories: no evidence, weak evidence, and strong evidence. As with EFI, these categories contain no information about the timing of infection.

We next assess the correspondence of EFI within mother/infant pairs. We compute the frequency of EFI category (none, weak, or strong for mothers versus infants), along with Pearson residuals from a model assuming independent association between mothers and infants. The residuals effectively rate the strength of association between the mother/infant pairs. We also test the dependence of our results on the specific threshold used to determine EFI strength. To do this, we rerun the above analysis using all individuals with either two or four detecting assays as sensitivity analyses to explore the impact of varying the EFI threshold (*Figure 7—figure supplement 1*).

We also assess the correspondence of EFI with clinical symptoms of pertussis and antibiotic use. Here we categorize symptoms as mild (cough or coryza) or moderate to severe (all other symptoms indicated by the modified Preziosi Scale [*Gill et al., 2016*]). We also categorize individuals according to the worst symptoms observed at any point during the study (*Préziosi and Halloran, 2003*). Likewise, participants are categorized according to whether antibiotics were used at any point during the study. This correspondence analysis was conducted separately for infants and mothers.

## Acknowledgements

We wish to thank the Lusaka lab team who generated the results for this analysis: Caitriona Murphy; Ruth Nkazwe; Chilufya Chikoti; and Baron Yankonde. The lab testing and subsequent analyses for this paper were supported by a grant from the National Institutes of Health/National Institute of Allergies and Infectious Diseases (R01AI133080). Funding for the initial SAMIPS study that allowed us to create the sample library itself was through the generous support of the Bill and Melinda Gates Foundation (OPP1105094).

## Additional information

### Funding

| Funder | Grant reference number | Author |
| --- | --- | --- |
| National Institutes of Health | R01AI133080 | Christopher J Gill |
| Bill and Melinda Gates Foundation | OPP1105094 | Christopher J Gill Pejman Rohani |

The funders had no role in study design, data collection and interpretation, or the decision to submit the work for publication.

### Author contributions

Christopher J Gill, Conceptualization, Resources, Supervision, Funding acquisition, Investigation, Visualization, Methodology, Writing - original draft, Project administration, Writing - review and editing; Christian E Gunning, Conceptualization, Resources, Data curation, Software, Formal analysis, Validation, Investigation, Visualization, Methodology, Writing - original draft, Project administration, Writing - review and editing; William B MacLeod, Conceptualization, Data curation, Methodology,

Writing - review and editing; Lawrence Mwananyanda, Supervision, Writing - review and editing; Donald M Thea, Writing - review and editing; Rachel C Pieciak, Project administration, Writing - review and editing; Geoffrey Kwenda, Zacharia Mupila, Investigation; Pejman Rohani, Conceptualization, Supervision, Funding acquisition, Methodology, Writing - original draft, Project administration, Writing - review and editing

## Author ORCIDs
Christopher J Gill  https://orcid.org/0000-0003-3353-0617
Christian E Gunning  https://orcid.org/0000-0001-6403-6553

## Ethics
Human subjects: The data for this analysis came from the Southern African Mother Infant Pertussis study (SAMIPS), which received ethical approval from the ERES converge IRB in Lusaka, and the IRB at Boston University Medical Center. The current analysis uses a de-identified data set at the request of the NIH for the R01 grant that supported this work. The current data sets contains no personally identifiable information.

## Decision letter and Author response
Decision letter https://doi.org/10.7554/eLife.65663.sa1
Author response https://doi.org/10.7554/eLife.65663.sa2

# Additional files
### Supplementary files
• Transparent reporting form

### Data availability
Data and analysis code are available at https://osf.io/yzxmv/.

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
