## [Decision Letter]

**Acceptance summary:**

In “Asymptomatic *Bordetella pertussis* infections in a longitudinal cohort of young African infants and their mothers”, the authors leverage unique longitudinal surveillance of a cohort of Zambian mother/infant pairs to investigate the role of asymptomatic infections on pertussis epidemiology. By analyzing these highly detailed data, they are able to show that the burden of pertussis is much higher in this population including frequent asymptomatic infections.

**Decision letter after peer review:**

Thank you for submitting your article "Asymptomatic *Bordetella pertussis* infections in a longitudinal cohort of young African infants and their mothers" for consideration by *eLife*. Your article has been reviewed by 3 peer reviewers, one of whom is a member of our Board of Reviewing Editors, and the evaluation has been overseen by a Senior Editor. The following individual involved in review of your submission has agreed to reveal their identity: Erik L Hewlett (Reviewer #3).

Essential revisions:

1) The role of vaccination: There was a limited discussion of vaccination that the reviewers felt was a limitation of the current manuscript. Multiple reviewers were interested with the role of vaccination in this population and how vaccination (particularly the wP vaccine) may impact CT values. Further, one reviewer suggested additional information about CT values (and particularly variability in these values) in relation to vaccination. Currently, there is limited investigation of how vaccination may have impacted these results, or possible implications of the generalizability of these results in populations with aP vaccine.

2) Additional investigation for EFI values that are able to provide more temporal information and the relationship with symptoms/antibiotic use. The authors have created an aggregate value which loses any temporality in these values (and their relationship with vaccination, symptoms, antibiotic use). The reviewers suggest performing a more detail analysis of these EFI measures that are able to capture more nuisance in aggregated CT values with many of the time-varying factors that the authors consider.

3) There are a number of format, language, and presentation issues (particularly those raised by Reviewer #3) that should be addressed.

*Reviewer #1 (Recommendations for the authors):*

In "Asymptomatic *Bordetella pertussis* infections in a longitudinal cohort of young African infants and their mothers", the authors analyze longitudinal data from a cohort in Zambia of infant/mother pairs to investigate the evidence for subclinical and asymptomatic infections in both pairs as well as the use of IS481 qPCR cycle threshold (CT) values in providing evidence for pertussis infection. Overall, the manuscript lacks substantial statistical support or clear evidence of some of the patterns they are stating and would require a substantial revision to justify their conclusions. The majority of the manuscript relies on 8 infant/mother pairs where they have evidence of pertussis infection and rely on the dense sampling to investigate infection dynamics. However, this is a very small sample size and further, based on the results displayed in Figure 1, it is not obvious that the data has a very pattern that warrant their assertions.

The main results and conclusions are highly reliant on details from eight mother/infant pairs. However, Figure 1 does not show a clear picture of the fade-in/fade-out. The authors go into great detail describing each of these 8 pairs, however based on the figure and text there does not appear to be clear evidence of an underlying pattern. While there are some instances with a combination of higher/lower IS481 CT values, it does not appear to have a clear pattern. For example, what are possible explanations for time periods between samples with evidence of IS481 and those without (such as pair A, C, D, E, F and H)? There also does not appear to be a clear pattern of symptoms in any of these samples (aside from having fewer symptoms in the mothers than infants). Further, it is not obvious how similar these observed (such as a mixture of times of high or low values often preceded or followed by times when IS481 was not detected) is similar to different to the rest of the cohort (in contrast to those who have a definitive positive NP sample during a symptomatic visit). The main results are primarily a descriptive analysis of these 8 mother/infant pairs with little statistical analyses or additional support.

The authors do not provide evidence or detail about what is known about the variability in IS481 CT values, amongst individuals, or over time, or pre/post vaccination. Without this information, it is not clear how informative some of this variability is versus how much variability in these values is expected. I think particularly in Figure 1, how many of the individuals have periods between times when IS481 evidence was observed when it was not observed, is concerning that these data (at this granular a level) are measuring true infection dynamics. Adding in additional information about the distribution and patterns of these values for the other cohort members would also provide valuable insight into how Figure 1 should be interpreted in this context. As it stands, the authors do not provide sufficient interpretation and evidence for having relevant infection arcs.

It appears that Figure 2A was created using only 8 data points (from the infant data values). If so, this level of extrapolation from such few data points does not provide enough evidence to support to the results in the text (particularly about evidence for fade-in/fade-out population-level dynamics). Also, in Figure 2, it is not clear to me the added value of Figure 2C and the main goal of this figure.

The authors have created a measure called, evidence for infection (EFI), which is a summary measure of their IS481 CT values across the study. However, it is not clear why the authors are only considering an aggregated (sum) value which loses any temporality or relationship with symptoms/antibiotic use. For example, the values may have been high earlier in the study, but symptoms were unrelated to that evidence for infection – or vice versa. This seems to be an important factor – were these possible undiagnosed, asymptomatic, or mild symptomatic pertussis infections? It is not clear why the authors only focus on a sum value for EFI versus other measures (such as multiple values above or below certain thresholds, etc.) to provide additional support and evidence for their results.

It is not clear why the authors have emphasized the novelty and large proportion of asymptomatic infections observed in these data. For example, there have been household studies of pertussis (see https://academic.oup.com/cid/article-abstract/70/1/152/5525423?redirectedFrom=PDF which performed a systematic review that included this topic) that have also found such evidence. While cross-sectional surveys may be commonly used in practice, it is not clear that there is no other type of study that provides any evidence for asymptomatic infections. Further, it is not clear why the authors refer to widespread asymptomatic pertussis when a large proportion of individuals with evidence for pertussis infection had symptoms. Would it not be undiagnosed pertussis if it is associated with clinical symptomatology?

*Reviewer #2 (Recommendations for the authors):*

Paragraph starting on line 218: the text refers to Figures 2C and 2D which I believe should be 3C and 3D.

*Reviewer #3 (Recommendations for the authors):*

This paper presents what I consider quite compelling data and discussion in support of the concept of asymptomatic infection with *B. pertussis*. The majority of concerns and questions that I have involve format, language and presentation.

• Although I am unable to locate it at the moment, I recall a very old paper in which NP swab cultures (prior to PCR assay) were taken from "healthy, asymptomatic individuals" in a population during an outbreak of pertussis. Culture-positive specimens were obtained from asymptomatic individuals, but all subjects either went on to become symptomatic or became culture negative in a relatively short period of time and were interpreted as reflecting subjects who were, indeed, previously immune, had been recently exposed and were in the early phase of clearing the organisms. I will continue to look for this publication and provide it, if I am successful.

• I gather from Gill et al. (CID, 2016) and this manuscript that all study subjects had been immunized and, appropriately, there was no "control" group. Was there, however, information on PCR or other results from mothers or infant in the population who were not part of the immunization study? Is asymptomatic infection with *B. pertussis* more likely to occur in a previously immunized subject?

• Page 9, line 225, cites Figure 2D as highlighting "…the strong association of EFI category within mother/infant pairs." I am not sure what that means and do not see a D-panel among the data in Figure 2.

• Also on page 9, lines 238 and 237, the authors state that a higher EFI "…represents a higher bacterial burden over time and thus yields a higher likelihood of symptomatic disease." This statement is not referenced and it is not clear on what basis it is made. I am not sure that we know that association.

• IS481 is present in high copy number in *B. pertussis*, but is also present in some B. bronchiseptica. Have there been any known infections of humans with B. bronchiseptica in this population, especially given that there is the possibility of significant animal exposure?

[Editors' note: further revisions were suggested prior to acceptance, as described below.]

Thank you for submitting your revised article "Asymptomatic *Bordetella pertussis* infections in a longitudinal cohort of young African infants and their mothers" for consideration by *eLife*. Your article has been re-evaluated by the initial reviewers, and the evaluation has been overseen by a Reviewing Editor and a Senior Editor.

What follows below is the Reviewing Editor's edited compilation of the essential and ancillary points provided by reviewers in their critiques and in their interaction post-review. Please submit a revised version that addresses these concerns directly. Although we expect that you will address these comments in your response letter, we also need to see the corresponding revision clearly marked in the text of the manuscript. Some of the reviewers' comments may seem to be simple queries or challenges that do not prompt revisions to the text. Please keep in mind, however, that readers may have the same perspective as the reviewers. Therefore, it is essential that you attempt to amend or expand the text to clarify the narrative accordingly.

Essential revisions:

1) We would just like some clarity on data availability and some additional details on where and how readers will be able to access the data, in particular the 1320 pairs with > 4 NP swabs.

2) The revised introduction (line 82), which states: "… where we use full-range IS481 CT values to show that qPCR signals of different intensities cluster in time, and summarize within-subject variations in signal intensity over time occur." This "occur" is left over from previous version and should be deleted.

3) In the last line of legend for figure 1. the authors state: "Note that, within a mother/infant pair, subject NP sample numbers match (with rare exceptions)." It is not clear how this information can be derived from either A or B (bar graph) of figure 1.

4) In the third line of the legend for figure 2, it is confusing to refer to the visits as "symptomatic" or not, in this case, the sentence ends "….during a symptomatic visit", which could be misread as asymptomatic. The visits are not "symptomatic" or not, they are visits in which the subject is experiencing symptoms or not.

---

## [Author Response]

Essential revisions:1) The role of vaccination: There was a limited discussion of vaccination that the reviewers felt was a limitation of the current manuscript. Multiple reviewers were interested with the role of vaccination in this population and how vaccination (particularly the wP vaccine) may impact CT values. Further, one reviewer suggested additional information about CT values (and particularly variability in these values) in relation to vaccination. Currently, there is limited investigation of how vaccination may have impacted these results, or possible implications of the generalizability of these results in populations with aP vaccine.

We agree that the role of vaccination in this study warrants further attention. As reviewer 3 notes, this study (necessarily) lacks a proper control. We recently published an analysis of adherence to the infant vaccine schedule in this cohort (see Gunning et al., BMJ Open 2020), where we found that non-adherence was surprisingly common despite our best efforts in the study. However, non-compliant infants also tended to be non- or minimally-participating infants who did not meet the necessary participation thresholds for this study (4 or more visits). We do, however, endeavor to address this question with additional analysis here. Overall, we find surprisingly little impact of prior vaccination on the presence of detecting assays (Figure 5). We qualify this with two additional observations. First, the ideal study design to assess an impact of this kind would have been a randomized controlled trial, and our study is not well-suited to identify any such effects. Second, we expect the impact of the wP vaccine used in this population on infection prevalence and severity to accrue in a non-linear fashion over time and with increasing doses, which is to say that such an effect might not be readily apparent until after dose 3. And since our study ended with the administration of dose 3, our window of observation ended prior to this period of particular interest. Therefore, we are reluctant to draw firm conclusions about the impact of vaccination on our results in the current analysis.

We also note that Chawama compound is broadly representative of populous urban slums worldwide, where primary care facilities are lacking and/or under-resourced, wP is the de facto standard, and facilities to diagnose and treat active pertussis infectious (e.g., with macrolides) are largely absent. As such, we believe our study is broadly generalizable to the substantial under-served and under-studied population of urban global poor.

2) Additional investigation for EFI values that are able to provide more temporal information and the relationship with symptoms/antibiotic use. The authors have created an aggregate value which loses any temporality in these values (and their relationship with vaccination, symptoms, antibiotic use). The reviewers suggest performing a more detail analysis of these EFI measures that are able to capture more nuisance in aggregated CT values with many of the time-varying factors that the authors consider.

We agree that the time-course of individuals within this cohort contains critical epidemiological information. Indeed, we believe the wealth and complexity of these data are sufficient to warrant a separate manuscript that compliments and extends this current manuscript with detailed time series analysis of study individuals, which we are currently drafting.

The work presented here focuses on high-level summaries of directly observed empirical measures. We have added two additional analyses that further explore the temporal structure of these data: A) the transition frequency of IS481 assays within individuals throughout the study (Figure 6), and B) the relationship between qPCR signal intensity and symptoms within clinic visits (Figure 8—figure supplement 1).

3) There are a number of format, language, and presentation issues (particularly those raised by Reviewer #3) that should be addressed.

Again we thank the reviewers for their careful reading and helpful suggestions. We respond to specific issues point-by-point, below. We note that we have modified and consolidated the qPCR signal intensity strata used now in Table 3 and Figures 3, 4, and 6. This change does not affect subsequent analyses involving EFI, and was made solely to clarify the presentation.

Reviewer #1 (Recommendations for the authors):In "Asymptomatic Bordetella pertussis infections in a longitudinal cohort of young African infants and their mothers", the authors analyze longitudinal data from a cohort in Zambia of infant/mother pairs to investigate the evidence for subclinical and asymptomatic infections in both pairs as well as the use of IS481 qPCR cycle threshold (CT) values in providing evidence for pertussis infection.Overall, the manuscript lacks substantial statistical support or clear evidence of some of the patterns they are stating and would require a substantial revision to justify their conclusions. The majority of the manuscript relies on 8 infant/mother pairs where they have evidence of pertussis infection and rely on the dense sampling to investigate infection dynamics. However, this is a very small sample size and further, based on the results displayed in Figure 1, it is not obvious that the data has a very pattern that warrant their assertions.The main results and conclusions are highly reliant on details from eight mother/infant pairs. However, Figure 1 does not show a clear picture of the fade-in/fade-out. The authors go into great detail describing each of these 8 pairs, however based on the figure and text there does not appear to be clear evidence of an underlying pattern.

As noted in the introduction, we begin our results with “a descriptive analysis of eight

8 mother/infant pairs where each symptomatic infant had definitive qPCR-based evidence of pertussis infection.” Our goal in this section is to use noteworthy examples to highlight salient epidemiological patterns, which we explore in further detail using data from the full cohort in subsequent sections. We note that the results presented in Figure 3 onwards in no way rely on any arguments and/or specific patterns described in Figure 2. In other words, the original eight pairs revealed several unanticipated findings (particularly the finding of repeated high CT values PCR findings in the mothers of a child with definite pertussis), that were intriguing and potentially relevant in terms of pertussis epidemiology. They are also unique – we have not seen any published time series data using qPCR in this way before. These early observations motivated us to conduct a more detailed and quantitative analysis of the cohort of >1,300 mother/infant pairs.

The sample size under consideration in the majority of the manuscript (i.e., all except for the above section) is 1,320 mother/infant pairs (2,640 subjects), as shown in Table 1 and 2. In the original submission, sample sizes were also clearly indicated in Figure 2B (assays per week), Figure 3B (subjects per group), Table 2 (subjects per group), Figure S1-2 (study profile), Figure S3 (NP samples per infant), and Table S1.

We have revised the panel order and axes labels of the current Figure 3 to more clearly illustrate the relationship between panels, and to clarify that the 6 example pairs shown in Figure 3A are unrelated to the 8 pairs shown in Figure 2. We hope this addresses any remaining confusion.

While there are some instances with a combination of higher/lower IS481 CT values, it does not appear to have a clear pattern. For example, what are possible explanations for time periods between samples with evidence of IS481 and those without (such as pair A, C, D, E, F and H)? There also does not appear to be a clear pattern of symptoms in any of these samples (aside from having fewer symptoms in the mothers than infants).

The ambiguity of these patterns played a role in guiding our analysis of the entire cohort, where we establish evidence for infection based on a preponderance of evidence from a large number of individuals.

Further, it is not obvious how similar these observed (such as a mixture of times of high or low values often preceded or followed by times when IS481 was not detected) is similar to different to the rest of the cohort (in contrast to those who have a definitive positive NP sample during a symptomatic visit). The main results are primarily a descriptive analysis of these 8 mother/infant pairs with little statistical analyses or additional support.

We strongly disagree with this characterization of our results, where we state that “In this analysis, we focus on the 1,320 pairs with ≥4 NP samples per subject (Figure S3)”. We believe the reviewer’s confusion may stem, in part, from a mis-interpretation of Figure 2 (below), along with our erroneous reference to Figure 3 (we incorrectly stated Figure 2, adding to the confusion). With this in mind, we have revised the previous Figure 2 (now Figure 3) in the interest of clarity, and more carefully described exactly what the points displayed in Figure 3 represent.

The authors do not provide evidence or detail about what is known about the variability in IS481 CT values, amongst individuals, or over time, or pre/post vaccination. Without this information, it is not clear how informative some of this variability is versus how much variability in these values is expected.

We agree that this is important information, and we have added figures and results summarizing the observed impact of vaccination on CT values (see essential revision 1, above), and the patterns of transitions of CT values across adjacent samples within individuals throughout the study (see essential revision 2). This latter analysis is now summarized in Figure 6, and shows a clear tendency for step-wise transitions over time. The implication is that the data present structure rather than random noise. This supports our overall contention that full-range CT values can provide meaningful insights into pertussis epidemiology. We also note that Figure 7A (previously Figure 3A) and Table 3 (previously Table S1) do indeed summarize the distribution of CT values, including variability amongst individuals. As noted above, we have also included an additional analysis summarizing the interdependence of CT value on both symptoms and antibiotics (Figure 8—figure supplement 1).

I think particularly in Figure 1, how many of the individuals have periods between times when IS481 evidence was observed when it was not observed, is concerning that these data (at this granular a level) are measuring true infection dynamics. Adding in additional information about the distribution and patterns of these values for the other cohort members would also provide valuable insight into how Figure 1 should be interpreted in this context.

We believe our previous comments concerning the relationship between the current Figure 2 (illustrative example) and the remaining figures (cohort analysis) addresses this comment.

As it stands, the authors do not provide sufficient interpretation and evidence for having relevant infection arcs.

We have revised the manuscript to clarify that infection arcs are observed in other studies and expected in infected individuals, rather than directly observed and/or quantified in this study.

It appears that Figure 2A was created using only 8 data points (from the infant data values). If so, this level of extrapolation from such few data points does not provide enough evidence to support to the results in the text (particularly about evidence for fade-in/fade-out population-level dynamics). Also, in Figure 2, it is not clear to me the added value of Figure 2C and the main goal of this figure.

We believe our previous comments have addressed this point. As noted, we have revised the current Figure 3 for clarity. Figure 3A and 3C are intended to demonstrate the structure of the cohort across the study period. We have revised the caption to clarify this point.

The authors have created a measure called, evidence for infection (EFI), which is a summary measure of their IS481 CT values across the study. However, it is not clear why the authors are only considering an aggregated (sum) value which loses any temporality or relationship with symptoms/antibiotic use. For example, the values may have been high earlier in the study, but symptoms were unrelated to that evidence for infection – or visa versa.

As noted in our response to essential revision #2, we believe that temporal patterns of CT values within subjects now described in Figure 6 deserve further detailed attention that is outside the scope of the current work. We believe the high-level empirical summaries presented here are strengthened by their reliance on a preponderance of evidence. In the current revision, we have also included additional analyses that we believe address some (if not all) of the reviewers concerns.

This seems to be an important factor – were these possible undiagnosed, asymptomatic, or mild symptomatic pertussis infections? It is not clear why the authors only focus on a sum value for EFI versus other measures (such as multiple values above or below certain thresholds, etc.) to provide additional support and evidence for their results.

Our approach seeks to use an objective statistical summary (geometric mean RCD proportion) to quantify the “signal” contained in IS481 assays within individuals across the course of the study. We note that, while both false positives and false negatives are likely in this study, the sample characteristics of the cohort mean that repeated false positives within individuals are unlikely based on chance alone. Further, a central aspect to our argument is that dichotomizing a continuous variable at an arbitrary threshold is reductive and unnecessarily introduces misclassification that reduces, rather than improves, statistical power.

It is not clear why the authors have emphasized the novelty and large proportion of asymptomatic infections observed in these data. For example, there have been household studies of pertussis (see https://academic.oup.com/cid/article-abstract/70/1/152/5525423?redirectedFrom=PDF which performed a systematic review that included this topic) that have also found such evidence.

We are aware of the paper above, which we had cited in the discussion. A key limitation of the referenced study is reliance on retrospective recall spanning many months. Since pertussis infections may be mild and non-specific, the fact that household contacts of an index case cannot recall a pertussis-like infection is consistent with asymptomatic infection, but far from definitive evidence. Moreover, the use of seroconversion as the measure of exposure is unreliable, since variations in antibody concentrations can be driven by a number of factors other than natural exposure.

While cross-sectional surveys may be commonly used in practice, it is not clear that there is no other type of study that provides any evidence for asymptomatic infections.

Our core argument is that it is impossible to know with certainty that a symptom-free patient with a detecting qPCR on Monday would not have become symptomatic if recontacted on Tuesday. By their nature, cross-sectional studies simply cannot parse asymptomatic from pre-symptomatic infections. To do that, one needs a longitudinal design, as reflected in the aforementioned longitudinal household contact studies. A key consideration addressed in the current work is the extent to which low and/or borderline CT values should be reinterpreted within the context of A) repeated sampling of individuals over time and B) epidemiological surveillance versus clinical diagnosis. We do not claim that our approach is the only one possible.

Further, it is not clear why the authors refer to widespread asymptomatic pertussis when a large proportion of individuals with evidence for pertussis infection had symptoms. Would it not be undiagnosed pertussis if it is associated with clinical symptomatology?

We have revised the text to highlight the significance of both asymptomatic and minimally symptomatic pertussis. As we describe both here and in Gill et al. 2016, only a handful of individuals meet the consensus criteria for clinical pertussis (Ct<35). In addition, qPCR results were not available to clinic staff in real-time. This, coupled with the relative absence of severe symptoms during study visits (especially in mothers), meant that only one study participant was diagnosed with pertussis at the time of their visit.

Reviewer #2 (Recommendations for the authors):Paragraph starting on line 218: the text refers to Figures 2C and 2D which I believe should be 3C and 3D.

Fixed, thank you.

Reviewer #3 (Recommendations for the authors):This paper presents what I consider quite compelling data and discussion in support of the concept of asymptomatic infection with B. pertussis. The majority of concerns and questions that I have involve format, language and presentation.• Although I am unable to locate it at the moment, I recall a very old paper in which NP swab cultures (prior to PCR assay) were taken from "healthy, asymptomatic individuals" in a population during an outbreak of pertussis. Culture-positive specimens were obtained from asymptomatic individuals, but all subjects either went on to become symptomatic or became culture negative in a relatively short period of time and were interpreted as reflecting subjects who were, indeed, previously immune, had been recently exposed and were in the early phase of clearing the organisms. I will continue to look for this publication and provide it, if I am successful.

From the reviewer’s description, we suspect that this may refer either to studies published by Long et al. or, more likely, the subsequent similar but much larger study by Deen et al. The references are provided below. The approach was similar in both cases: starting with index cases of children with clinical pertussis disease, household contacts were assessed at about a year after and seroconversion rates documented. Those cases were assumed to represent secondary spread from the index cases and a significant fraction of those who seroconverted could not recall a pertussis-like syndrome. On this basis, they were presumptive examples of asymptomatic pertussis.

We agree that these are relevant citations for our manuscript and have added them to the paper along with further narrative explaining their contextual relevance.

Long SS, Welkon CJ, Clark JL. Widespread silent transmission of pertussis in families: antibody correlates of infection and symptomatology. *The Journal of infectious diseases.* Mar 1990;161(3):480-486.

Deen JL, Mink CA, Cherry JD, et al. Household contact study of *Bordetella pertussis* infections. *Clinical infectious diseases : an official publication of the Infectious Diseases Society of America.* Nov 1995;21(5):1211-1219.

• I gather from Gill et al. (CID, 2016) and this manuscript that all study subjects had been immunized and, appropriately, there was no "control" group. Was there, however, information on PCR or other results from mothers or infant in the population who were not part of the immunization study? Is asymptomatic infection with B. pertussis more likely to occur in a previously immunized subject?

This is largely correct in that we supported the infants in the cohort to receive their scheduled vaccines on time. This was more or less successful, and the ‘less successful’ proportion are of obvious interest because they provide a rough counterfactual. We recently published our results examining the adherence to the infant vaccine schedule (see Gunning et al., BMJ Open 2020 – note this was published after our initial submission, and has now been updated in the bibliography accordingly) where we found that non-adherence was surprisingly common despite our best efforts in the study. However, non-compliant infants also tended to be non- or minimally-participating infants who did not meet the necessary participation thresholds for this study (4 or more visits).

As the reviewer notes, we lack a proper control group. As an urban slum, epidemiological and/or clinical records from Chawama compound are sorely lacking. This absence of evidence is common in populous urban slums worldwide, which we believe adds to the significance and impact of our study, but does sharply limit the availability of contextual information.

We have, however, endeavored to address this question with additional analysis exploring the relationship between vaccination status and IS481 detection probability in both mothers and infants (Figure 5). Please also see our responses to Essential Revision 1.

• Page 9, line 225, cites Figure 2D as highlighting "…the strong association of EFI category within mother/infant pairs." I am not sure what that means and do not see a D-panel among the data in Figure 2.

We apologize for creating the confusion. The references to Figure 2 here should have referred to Figure 3 (now Figure 7). This was a typo from a previous draft, and has been corrected in the current paper. Thus, in the revised submission, Figure 7C shows EFI scores for mothers and infants, while Figure 7D shows the strong positive association between these scores (within mother/infant pairs): when mother has no EFI, that tends to be true of her baby; when mother has a strong EFI, the babies tend to be the same – and vice versa. The argument we advance with these data is that the EFI scores are not varying independently or randomly. Rather, mothers’ EFI mom predicts infants’ EFI. We submit that this strong positive association within mother/infant pairs demonstrates both

• Also on page 9, lines 238 and 237, the authors state that a higher EFI "…represents a higher bacterial burden over time and thus yields a higher likelihood of symptomatic disease." This statement is not referenced and it is not clear on what basis it is made. I am not sure that we know that association.

We can see why the reviewer found this section unclear. Our formulation of EFI is a proxy for average signal intensity (CT value) across the course of the study. As such, higher EFIs are associated with lower CT values (on average, across on one or more detecting assays), thus indicating higher observed pathogen burden across the course of the study. We have endeavored to clarify this section: to better explain the connection between EFI and CT values, and to cite evidence linking CT values to pathogen burden (see L288-292).

• IS481 is present in high copy number in B. pertussis, but is also present in some B. bronchiseptica. Have there been any known infections of humans with B. bronchiseptica in this population, especially given that there is the possibility of significant animal exposure?

It is an excellent question, but we are unaware of work on *B. bronchiseptica* in Zambia. We accept the reviewer’s point that there could be some misclassification of individual test results on this basis. We cannot quantify this precisely in the absence of contemporaneous *B. bronchiseptica* data, but in other publications this pathogen occurs at a low frequency compared with the human adapted *B. pertussis*. This could introduce a degree of misclassification in our analysis, which would tend to bias our results towards the null, rendering them more conservative. We have addressed this in the limitations section (line 296 in the original paper), and added a reference to the bibliography (Van den Bossche et al., 2013).

[Editors' note: further revisions were suggested prior to acceptance, as described below.]

Essential revisions:1) We would just like some clarity on data availability and some additional details on where and how readers will be able to access the data, in particular the 1320 pairs with > 4 NP swabs.

We have updated our data availability statement, and added included R code per correspondence with journal staff.

2) The revised introduction (line 82), which states: "… where we use full-range IS481 CT values to show that qPCR signals of different intensities cluster in time, and summarize within-subject variations in signal intensity over time occur." This "occur" is left over from previous version and should be deleted.

Fixed.

3) In the last line of legend for figure 1. the authors state: "Note that, within a mother/infant pair, subject NP sample numbers match (with rare exceptions)." It is not clear how this information can be derived from either A or B (bar graph) of figure 1.

This last sentence is simply a statement of fact that aids in interpretation of figures 1B (rather than following from this figure). We have edited this sentence for clarity. We note that we asked the editorial office if a specific change was requested here. We were informed by Sam on 9 April that “I have passed them on to the editors to see if they can provide any further clarification on what changes they are requesting”. We have not received further correspondence. We believe this is a minor point, and hope that our edits and the availability of our raw data allay any lingering concerns here.

4) In the third line of the legend for figure 2, it is confusing to refer to the visits as "symptomatic" or not, in this case, the sentence ends "….during a symptomatic visit", which could be misread as asymptomatic. The visits are not "symptomatic" or not, they are visits in which the subject is experiencing symptoms or not.

The text of the requested change appears to be Line 120 (4th para of results), which we have updated. The Figure 2 legend refers to symptomatic infants, which we presume is not problematic.